# Ancient Fennoscandian genomes reveal origin and spread of Siberian ancestry in Europe

Thiseas C. Lamnidis[1], Kerttu Majander[1,2,3], Choongwon Jeong[1,4], Elina Salmela [1,3], Anna Wessman[5], Vyacheslav Moiseyev[6], Valery Khartanovich[6], Oleg Balanovsky[7,8,9], Matthias Ongyerth[10], Antje Weihmann[10], Antti Sajantila[11], Janet Kelso [10], Svante Pääbo[10], Päivi Onkamo[3,12], Wolfgang Haak[1], Johannes Krause [1] & Stephan Schiffels [1]

European population history has been shaped by migrations of people, and their subsequent admixture. Recently, ancient DNA has brought new insights into European migration events linked to the advent of agriculture, and possibly to the spread of Indo-European languages. However, little is known about the ancient population history of north-eastern Europe, in particular about populations speaking Uralic languages, such as Finns and Saami. Here we analyse ancient genomic data from 11 individuals from Finland and north-western Russia. We show that the genetic makeup of northern Europe was shaped by migrations from Siberia that began at least 3500 years ago. This Siberian ancestry was subsequently admixed into many modern populations in the region, particularly into populations speaking Uralic languages today. Additionally, we show that ancestors of modern Saami inhabited a larger territory during the Iron Age, which adds to the historical and linguistic information about the population history of Finland.

[1] Department of Archaeogenetics, Max Planck Institute for the Science of Human History, 07745 Jena, Germany. [2] Institute for Archaeological Sciences, Archaeo- and Palaeogenetics, University of Tübingen, 72070 Tübingen, Germany. [3] Department of Biosciences, University of Helsinki, PL 56 (Viikinkaari 9), 00014 Helsinki, Finland. [4] The Eurasia3angle Project, Max Planck Institute for the Science of Human History, 07745 Jena, Germany. [5] Department of Cultures, Archaeology, University of Helsinki, PL 59 (Unioninkatu 38), 00014 Helsinki, Finland. [6] Peter the Great Museum of Anthropology and Ethnography (Kunstkamera), Russian Academy of Sciences, University Embankment, 3, Saint Petersburg 199034, Russia. [7] Vavilov Institute of General Genetics, Ulitsa Gubkina, 3, Moscow 117971, Russia. [8] Research Centre for Medical Genetics, Moskvorech'ye Ulitsa, 1, Moscow 115478, Russia. [9] Biobank of North Eurasia, Kotlyakovskaya Ulitsa, 3 строение 12, Moscow 115201, Russia. [10] Max Planck Institute for Evolutionary Anthropology, Deutscher Pl. 6, 04103 Leipzig, Germany. [11] Department of Forensic Medicine, University of Helsinki, PL 40 (Kytösuontie 11), Helsinki 00014, Finland. [12] Department of Biology, University of Turku, Turku 20014, Finland. These authors contributed equally to this work: Thiseas C. Lamnidis, Kerttu Majander. These authors jointly supervised this work: Päivi Onkamo, Wolfgang Haak, Johannes Krause, Stephan Schiffels. Correspondence and requests for materials should be addressed to P.O. (email: paivi.onkamo@helsinki.fi) or to W.H. (email: haak@shh.mpg.de) or to S.S. (email: schiffels@shh.mpg.de)

The genetic structure of Europeans today is the result of several layers of migration and subsequent admixture. The incoming source populations no longer exist in unadmixed form, but have been identified using ancient DNA in several studies over the last few years[1–8]. Broadly, present-day Europeans have ancestors in three deeply diverged source populations: European hunter-gatherers who settled the continent in the Upper Paleolithic, Europe's first farmers who expanded from Anatolia across Europe in the early Neolithic starting around 8000 years ago, and groups from the Pontic Steppe that arrived in Europe during the final Neolithic and early Bronze Age ~ 4500 years ago. As a consequence, most Europeans can be modelled as a mixture of these three ancestral populations[3].

This model, however, does not fit well for present-day populations from north-eastern Europe such as Saami, Russians, Mordovians, Chuvash, Estonians, Hungarians, and Finns: they carry additional ancestry seen as increased allele sharing with modern East Asian populations[1,3,9,10]. The origin and timing of this East Asian-related contribution is unknown. Modern Finns are known to possess a distinct genetic structure among today's European populations[9,11,12], and the country's geographical location at the crossroads of eastern and western influences introduces a unique opportunity to investigate the migratory past of north-east Europe. Furthermore, the early migrations and genetic origins of the Saami people in relation to the Finnish population call for a closer investigation. Here, the early-Metal-Age, Iron-Age, and historical burials analysed provide a suitable time-transect to ascertain the timing of the arrival of the deeply rooted Siberian genetic ancestry, and a frame of reference for investigating linguistic diversity in the region today.

The population history of Finland is subject to an ongoing discussion, especially concerning the status of the Saami as the earlier inhabitants of Finland, compared to Finns. The archaeological record proves human presence in Finland since 9000 BC[13]. Over the millennia, people from Scandinavia, the northeast Baltics, and modern-day Russia have left evidence of their material cultures in Finland[14,15]. The Finno-Ugric branch of the Uralic language family, to which both Saami and Finnish languages belong, has diverged from other Uralic languages no earlier than 4000–5000 years ago, when Finland was already inhabited by speakers of a language today unknown. Linguistic evidence shows that Saami languages were spoken in Finland prior to the arrival of the early Finnish language and have dominated the whole of the Finnish region before 1000 CE[16–18]. Particularly, southern Ostrobothnia, where Levänluhta is located, has been suggested through place names to harbour a southern Saami dialect until the late first millennium[19], when early Finnish took over as the dominant language[20]. Historical sources note Lapps living in the parishes of central Finland still in the 1500s[21]. It is, however, unclear whether all of them spoke Saami, or if some of them were Finns who had changed their subsistence strategy from agriculture to hunting and fishing. There are also documents of intermarriage, although many of the indigenous people retreated to the north (see ref. [22] and references therein). Ancestors of present-day Finnish speakers possibly migrated from northern Estonia, to which Finns still remain linguistically close, and displaced but also admixed with the local population of Finland, the likely ancestors of today's Saami speakers[23].

In this study, we present new genome-wide data from Finland and the Russian Kola Peninsula, from 11 individuals who lived between 3500 and 200 years ago (and 4 more ancient genomes with very low coverage). In addition, we present a new high-coverage whole genome from a modern Saami individual for whom genotyping data was previously published[1]. Our results suggest that a new genetic component with strong Siberian affinity first arrived in Europe at least 3500 years ago, as observed in our oldest analysed individuals from northern Russia. These results describe the gene pool of modern north-eastern Europeans in general, and of the speakers of Uralic languages in particular, as the result of multiple admixture events between Eastern and Western sources since the first appearance of this ancestry component. Additionally, we gain further insights into the genetic history of the Saami in Finland, by showing that during the Iron Age, close genetic relatives of modern Saami lived in an area much further south than their current geographic range.

## Results

**Sample information and archaeological background**. The ancient individuals analysed in this study come from three time periods (Table 1, Fig. 1, Supplementary Note 1). The six early Metal Age individuals were obtained from an archaeological site at Bolshoy Oleni Ostrov in the Murmansk Region on the Kola Peninsula (Bolshoy from here on). The site has been radiocarbon dated to 1610-1436 calBCE (see Supplementary Note 1) and the mitochondrial DNA HVR-I haplotypes from these six individuals have been previously reported[24]. Today, the region is inhabited by Saami. Seven individuals stem from excavations in Levänluhta, a lake burial in Isokyrö, Finland. Artefacts from the site have been dated to the Finnish Iron Age (300–800 CE)[25,26]. Today, the inhabitants of the area speak Finnish and Swedish. Two individuals were obtained from the 18–19th century Saami cemetery of Chalmny Varre on the Russian Kola Peninsula. The cemetery and the surrounding area were abandoned in the 1960s because of planned industrial constructions, and later became the subject of archaeological excavations. In addition, we sequenced the whole genome of a modern Saami individual to 17.5-fold coverage, for whom genotyping data has previously been published[1].

The sampling and subsequent processing of the ancient human remains was done in dedicated clean-room facilities (Methods). A SNP-capture approach targeting a set of 1,237,207 single nucleotide polymorphisms (SNPs) was used to enrich ancient-DNA libraries for human DNA[4]. The sequenced DNA fragments were mapped to the human reference genome, and pseudohaploid genotypes were called based on a random read covering each targeted SNP (see Methods). To ensure the ancient origin of our samples, and the reliability of the data produced, we implemented multiple quality controls. First, we confirmed the deamination patterns at the terminal bases of DNA reads being characteristic of ancient DNA (Supplementary Table 1). Second, we estimated potential contamination rates through heterozygosity on the single-copy X chromosome for male individuals (all below 1.6%)[27] (Table 1, see Supplementary Figure 1 for sex determination). Third, we carried out supervised genetic clustering using ADMIXTURE[28], using six divergent populations as defined clusters, to identify genetic dissimilarities and possible contamination from distantly related sources (Supplementary Figure 2, Supplementary Note 2). Fourth, we calculated mitochondrial contamination for all ancient individuals using ContamMix[29]. Finally, for individuals with sufficient SNP coverage, we carried out principal component analysis (PCA), projecting damage-filtered and non-filtered versions of each individual, to show that the different datasets cluster together regardless of the damage-filtering[30] (see Supplementary Figure 3 and Methods). Eleven ancient individuals passed those quality checks, while four individuals from Levänluhta were excluded from further analyses, due to low SNP coverage (<15,000 SNPs). We merged the data from these 15 individuals with that of 3333 published present-day individuals genotyped on the Affymetrix Human Origins platform and 538 ancient individuals sequenced using a mixture of DNA capture and shotgun sequencing (Supplementary Data 1)[1–4,8,31,32].

**Table 1 Sample information**

| Individual ID | Site/ location | Date | Population label | Genetic Sex | # SNPs overlap with Human Origins | Avg. coverage on target (1240 K MapQ ≥ 30) | Nuclear contamination | mtDNA contamination | mtDNA, Y haplotypes |
|---|---|---|---|---|---|---|---|---|---|
| BOO001 | Bolshoy Oleni Ostrov, Murmansk, Russia | 3473 ± 87 calBP | Bolshoy | F | 347,709 | 1.22 | N/A | 0.001 (0.000–0.007) | U4a1 |
| BOO002 | | | Bolshoy | M | 403,994 | 0.81 | 0.004 ± 0.003 | 0.000 (0.000–0.002) | Z1a1a, N1c1a1a |
| BOO003 | | | Bolshoy | F | 300,598 | 1.09 | N/A | 0.016 (0.010–0.024) | T2d1b1 |
| BOO004 | | | Bolshoy | M | 342,582 | 2.92 | 0.002 ± 0.001 | 0.003 (0.000–0.059) | C4b, N1c1a1a |
| BOO005 | | | Bolshoy | F | 347,042 | 2.88 | N/A | 0.005 (0.001–0.026) | U5a1d |
| BOO006 | | | Bolshoy | F | 390,835 | 1.08 | N/A | 0.008 (0.004–0.014) | D4e4 |
| CHV001 | Chalmny- Varre, Murmansk, Russia | 18–19th cent CE | Chalmny Varre | F | 426,702 | 1.42 | N/A | 0.001 (0.000–0.003) | U5b1b1a3 |
| CHV002 | | | Chalmny Varre | M | 215,228 | 0.47 | 0.016 ± 0.011 | 0.000 (0.000–0.003) | V7a1, I2a1 |
| JK1963 | Levänluhta, Isokyrö, Finland | 300–800 CE | N/A | f | 10,492 | 0.02 | N/A | 0.004 (0.001–0.013) | N/A |
| JK1967 | | | N/A | f | 5267 | 0.01 | N/A | 0.000 (0.000–0.009) | N/A |
| JK1968 | | | Levänluhta | F | 207,076 | 0.47 | N/A | 0.004 (0.001–0.065) | U5a1a1a'b'n |
| JK1970 | | | Levänluhta | F | 194,764 | 0.44 | N/A | 0.023 (0.005–0.129) | U5a1a1 |
| JK2065 | | | Levänluhta_B | F | 133,224 | 0.29 | N/A | 0.028 (0.012–0.064) | K1a4a1b |
| JK2066 | | | N/A | ? | 4045 | 0.01 | N/A | 0.002 (0.001–0.007) | N/A |
| JK2067 | | | N/A | ? | 2356 | 0.01 | N/A | 0.010 (0.002–0.032) | N/A |
| Saami001 | Finland | Modern | Modern Saami | M | 593,094 | 6.32 | N/A | N/A | U5b1b1a1, I1a1b3a1 |

Summary information for all individuals for which we report genomic data in this study. A radiocarbon date is given for the Bolshoy samples, as described in Supplementary Text 1. Other dates are context-based, as described in Supplementary Text 1. Population labels for individuals of low coverage are shown as N/A. Genetic sex is determined as described in Methods, and lowercase letters denote probable genetic sex for low-coverage individuals. Question marks denote undefined genetic sex due to low coverage. Mitochondrial contamination estimates are given as posterior mode and 95% posterior intervals in brackets, as provided by ContamMix[70]

**Eastern genetic affinities in Northern Europe**. To investigate the genetic affinities of the sampled individuals, we projected them onto principal components (PC) computed from 1320 modern European and Asian individuals (Fig. 2a, Supplementary Figures 3a, b for a version focusing on West Eurasia). As expected, PC1 separates East Asian from West Eurasian populations. Within each continental group, genetic variability is spread across PC2: The East Asian genetic cline contains populations between the Siberian Nganasan (Uralic speakers) and Yukagirs at one end, and the Ami and Atayal from Taiwan at the other end. The West Eurasian cline along PC2 spans from the Bedouins on the Arabian Peninsula to north-eastern Europeans including Lithuanians, Norwegians and Finns. Between these two main Eurasian clines exist multiple clines, spanning between West and East Eurasians. These clines are likely the result of admixture events and population movements between East and West Eurasia. Most relevant to the populations analysed here is the admixture cline between north-eastern Europe and the North Siberian Nganasan, including mostly Uralic-speaking populations in our dataset (marked in light purple in Fig. 2a).

Ten of the eleven ancient individuals from this study fall on this Uralic cline, with the exception of one individual from Levänluhta (ID JK2065, here named Levänluhta_B), who instead is projected closer to modern Lithuanian, Norwegian and Icelandic populations. Specifically, two Levänluhta individuals and the two historical Saami from Russia are projected very close to the two previously published modern Saami (Saami.DG)[32] and the new Saami shotgun genome generated in this study (as well as the previously published genome of the same individual, here labelled Saami (WGA)[1]), suggesting genetic continuity in the north from the Iron Age to modern-day Saami populations. In contrast, the six ancient individuals from Bolshoy are projected much further towards East Asian populations, and fall to an intermediate position along the Uralic cline and close to modern-day Mansi.

Unsupervised genetic clustering analysis as implemented in the ADMIXTURE[28] program suggests a similar profile to the PCA: north-eastern European populations harbour a Siberian genetic component (light purple) maximized in the Nganasan (Fig. 2b, see Supplementary Figure 4a for results over multiple K values). The hunter-gatherer genetic ancestry in Europeans (blue) is maximized in European Upper Palaeolithic and Mesolithic hunter-gatherers, including the 8000-year-old Western European hunter-gatherers from Hungary and Spain (WHG), the 8000-year-old Scandinavian hunter-gatherers from Motala (SHG) and the Narva and Kunda individuals from the Baltics. An ancestry component associated with Europe's first farmers (orange) is maximized in Early Neolithic Europeans associated with the LBK (from German: Linearbandkeramik). The steppe ancestry component within modern Europeans (green), which is associated with the Yamnaya population, is maximised in ancient Iranian populations and to a lesser extent Caucasus hunter-gatherers (CHG). This ancestry component is also present in modern Armenians from the Caucasus, Bedouins from the Arabian Peninsula and South Asian populations. Within modern Europeans, the Siberian genetic component (light purple) is maximized in the Mari and Saami, and can also be seen in similar proportions in the historical Saami from Chalmny Varre and in two of the Levänluhta individuals. The third Levänluhta individual (Levänluhta_B), however, lacks this Siberian

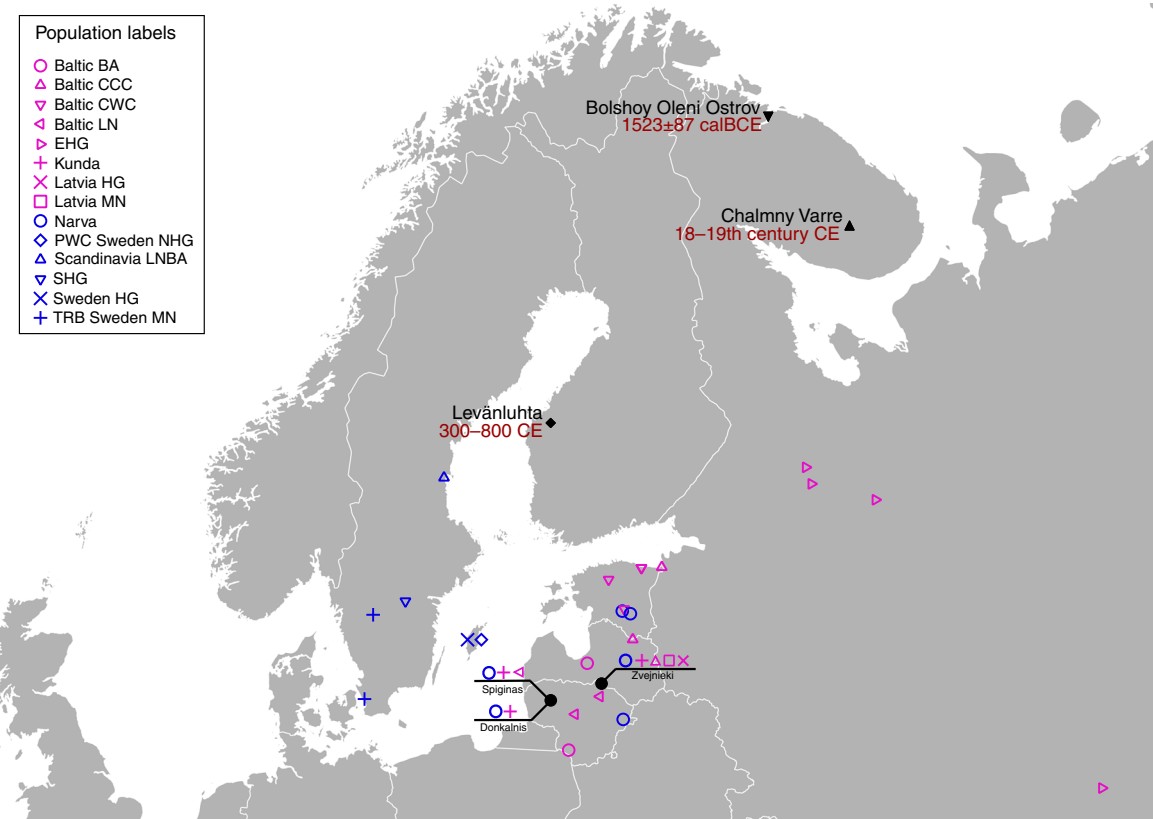

**Fig. 1** Location and age of archaeological sites used in this study. The location of other sites relevant to this study is also shown. Markers used correspond to the ones used in Fig. 2. Map generated with QGIS 2.18.19 (http://www.qgis.org/) using the Natural Earth country boundary dataset (http://www.naturalearthdata.com) for the basemap. Source data are provided as a Source Data file

component. The six ancient individuals from Bolshoy show substantially higher proportions of the Siberian component: it comprises about half of their ancestry (42.3–58.2%), whereas the older Mesolithic individuals from Motala (SHG) do not possess it at all. The Native-American-related ancestry seen in the EHG and Bolshoy corresponds to a previously reported affinity towards Ancient North Eurasians (ANE)[2,33] contributing genes to both Native Americans and West Eurasians. ANE ancestry also comprises part of the ancestry of Nganasans[2].

Interestingly, results from uniparentally-inherited markers (mtDNA and Y chromosome) as well as certain phenotypic SNPs also show Siberian signals in Bolshoy: mtDNA haplogroups Z1, C4 and D4, common in modern Siberia[24,34,35] are represented by the individuals BOO002, BOO004 and BOO006, respectively (confirming previous findings[24]), whereas the Y-chromosomal haplotype N1c1a1a (N-L392) is represented by the individuals BOO002 and BOO004. Haplogroup N1c, to which this haplotype belongs, is the major Y-chromosomal lineage in modern north-east Europe and European Russia. It is especially prevalent in Uralic speakers, comprising for example as much as 54% of eastern Finnish male lineages today[36]. Notably, this is the earliest known occurrence of Y-haplogroup N1c in Fennoscandia. Additionally, within the Bolshoy population, we observe the derived allele of rs3827760 in the *EDAR* gene, which is found in near-fixation in East Asian and Native American populations today, but is extremely rare elsewhere[37], and has been linked to phenotypes related to tooth shape[38] and hair morphology[39] (Supplementary Data 2). Scandinavian hunter-gatherers from Motala in Sweden have also been found to carry haplotypes associated with this allele[4]. Finally, in the Bolshoy individuals we also see high frequencies of haplotypes associated with diets rich in poly-unsaturated fatty acids, in the *FADS* genes[4,40,41].

**The arrival of Siberian ancestry in Europe.** We formally tested for admixture in north-eastern Europe by calculating $f_3$(Test; Siberian source, European source) statistics. We used several Uralic-speaking populations—Estonians, Saami, Finnish, Mordovians and Hungarians—and Russians as Test populations. Significantly negative $f_3$ values correspond to the Test population being admixed between populations related to the two source populations[42]. We used multiple European and Siberian sources to capture differences in ancestral composition among proxy populations: As proxies for the Siberian source we used Bolshoy, Mansi and Nganasan, and for the European source we used modern Icelandic, Norwegian, Lithuanian and French. Our results show that all of the test populations are indeed admixed, with the most negative values arising when Nganasan are used as the Siberian source (Supplementary Data 3). Among the European sources, Lithuanians gave the most negative results for Estonians, Russians and Mordovians. For modern Hungarians, the European source giving most negative results was French, while both Bolshoy and Nganasan gave equally negative results when used as the Siberian source. With Finns as test, modern Icelanders were the European source giving most negative statistics. Finally, Icelanders and Nganasan used as the European and Siberian sources, respectively, yielded the most negative result for the present-day Saami as a Test. This result is still non-significantly negative, either due to the low number of modern Saami individuals in our dataset ($n = 3$), or due to post-admixture drift in modern Saami. A high degree of population-specific drift can affect $f_3$-statistics and result in less negative and even positive values[42]. Indeed, post-admixture drift would correlate well with the suggested founder effect[43] in Saami. To further test differential relatedness with Nganasan in European populations and in the ancient individuals in this study, we

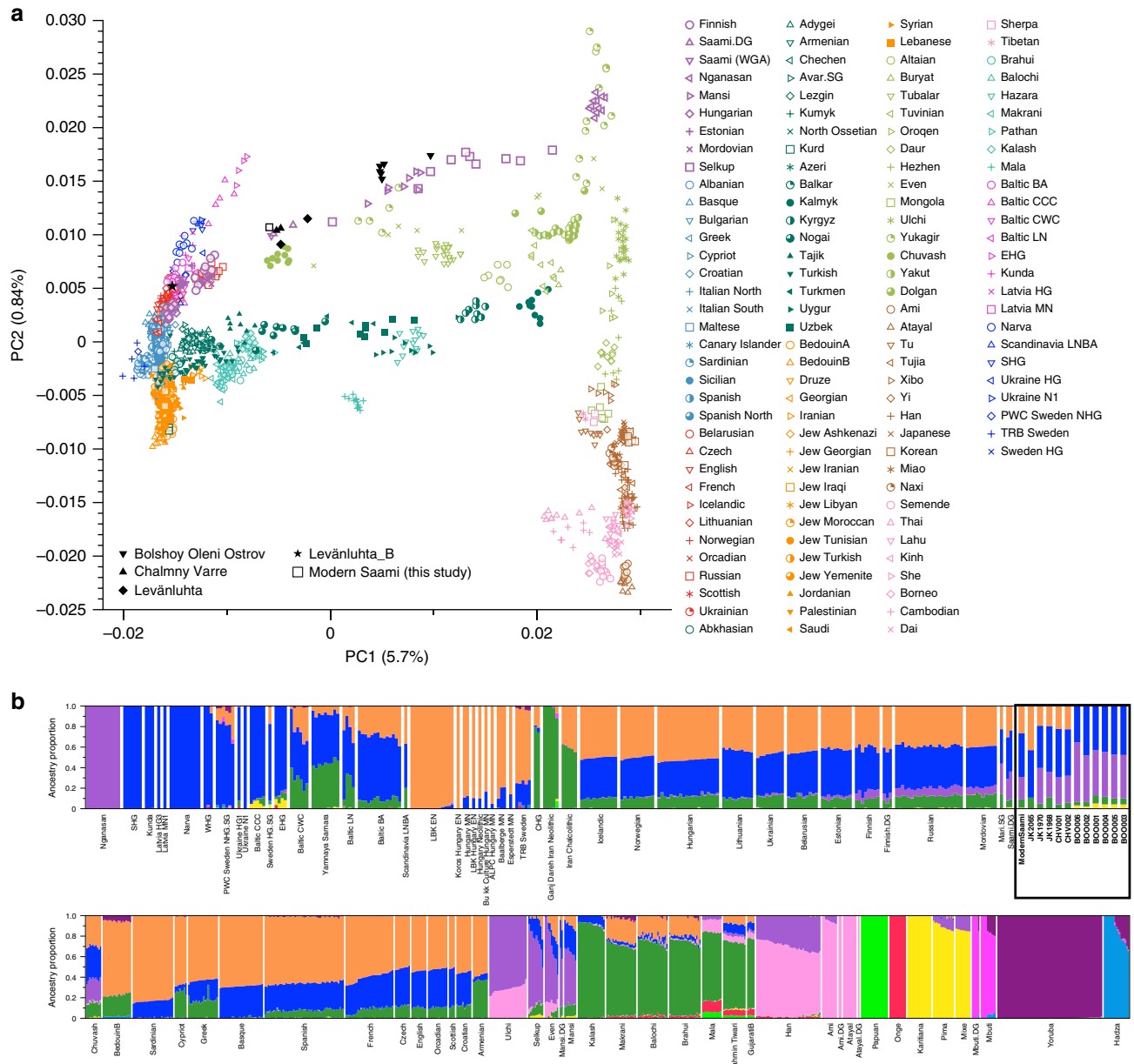

**Fig. 2** PCA and ADMIXTURE analysis. **a** PCA plot of 113 Modern Eurasian populations, with individuals from this study and other relevant ancient genomes projected on the principal components, using the "shrinkmode: YES" option. Uralic-speaking populations are highlighted in dark purple. PCA of Europe can be found in Supplementary Figure 3. **b** Plot of ADMIXTURE (K = 11) results containing worldwide populations. Ancient individuals from this study (black box) are represented by thicker bars and shown in bold. For a figure of ADMIXTURE results over multiple K values see Supplementary Figure 4a. Source data are provided as a Source Data file

calculated $f_4$(Mbuti, Nganasan; Lithuanian, Test) (Fig. 3). Consistent with $f_3$-statistics above, all the ancient individuals and modern Finns, Saami, Mordovians and Russians show excess allele sharing with Nganasan when used as Test populations. Of all Uralic-speaking populations in Europe, Hungarians are the only population that shows no evidence of excess allele sharing with Nganasan compared to that of Lithuanians, consistent with their distinct population history from other Uralic populations as evidenced by historical sources (see ref. [44] and references therein).

We further estimated the genetic composition in these populations using qpAdm[3]. All ancient and modern individuals from the Baltics, Finland and Russia were successfully modelled as a mixture of five lines of ancestry, represented by eastern Mesolithic hunter-gatherers (EHG, from Karelia), Yamnaya from Samara, LBK from the early European Neolithic, western

Mesolithic hunter-gatherers (WHG, from Spain, Luxembourg and Hungary), and Nganasan, or subsets of those five (Supplementary Data 4). In contrast to previous models for European populations using three streams of ancestry[2,3], we found that some populations modelled here require two additional components: a component related to modern Nganasans, as discussed above, and additional EHG ancestry, not explained by Yamnaya (who have been shown to contain large amounts of EHG ancestry themselves[3]). Indeed, the six Bolshoy individuals have substantial amounts of EHG but no Yamnaya ancestry. We find that Nganasan-related ancestry is significantly present in all of our ancient samples except for Levänluhta_B, and in many modern, mainly Uralic-speaking populations. The 3500-year-old ancient individuals from Bolshoy represent the highest proportion of Siberian Nganasan-related ancestry seen in this

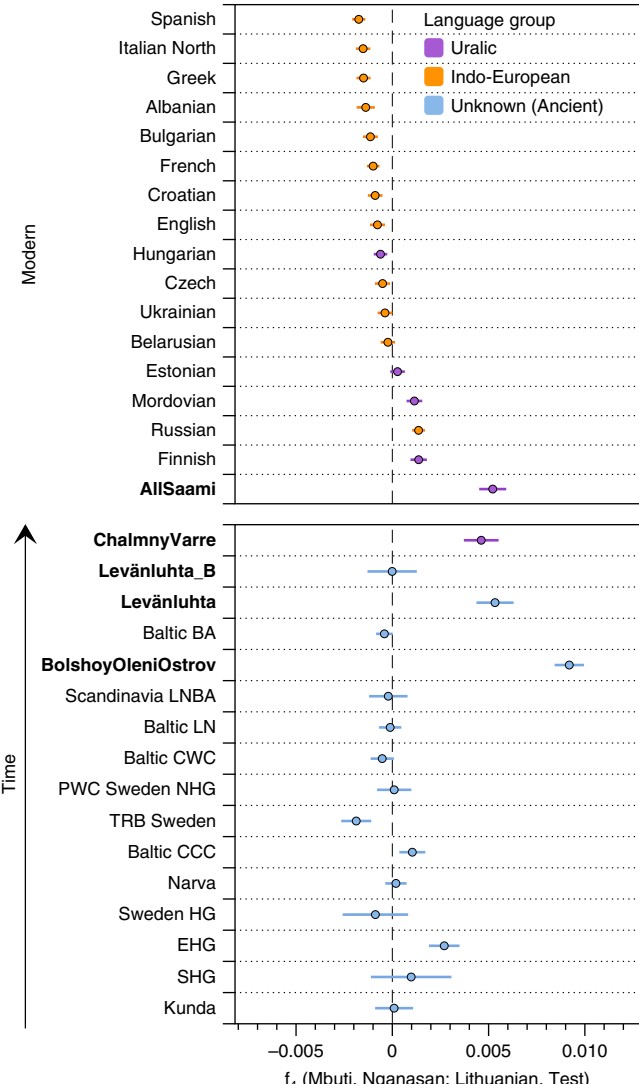

**Fig. 3** Calculated $f_4$ (Mbuti, Nganasan; Lithuanian, Test). Modern populations are sorted by $f_4$ magnitude; ancient populations are sorted through time. Populations are coloured based on language family, with Ancient individuals with unknown language shown in blue. "AllSaami" refers to a grouping including the 2 individuals from the SGDP (Saami (SGDP)) and the high-coverage modern Saami shotgun genome in this study (Modern Saami). Individuals from this study are indicated by labels in bold. Error bars represent 3 standard errors, to indicate significant difference from 0. Source data are provided as a Source Data file

Bolshoy works as source in some ancient individuals, but not for modern Uralic speakers (see Supplementary Data 4).

As shown by these multiple lines of evidence, the pattern of genetic ancestry observed in north-eastern Europe is the result of admixture between populations from Siberia and populations from Europe. To obtain a relative date of this admixture, and as an independent line of evidence thereof, we used admixture linkage disequilibrium decay, as implemented in ALDER[46]. Our ALDER admixture estimate for Bolshoy, using Nganasan and EHG as admixture sources, dates only 17 generations ago. Based on the radiocarbon date for Bolshoy and its uncertainty, and assuming a generation time of 29 years[47], we estimate the time of introduction of the Siberian Nganasan-related ancestry in Bolshoy to be 3977 (± 77) years before present (yBP) (Fig. 5b). Estimates obtained using Nganasans and Lithuanians as source populations provided a similar estimate (Supplementary Figure 5 for LD decay plots for multiple populations using Lithuanian and Nganasan as sources.). ALDER provides a relative date estimate for a single-pulse admixture event in generations. When multiple admixture events have occurred, such a single estimate should be interpreted as a (non-arithmetic) average of those events[46,48]. Therefore the admixture date estimate for Bolshoy does not preclude earlier admixture events bringing Siberian Nganasan-related ancestry into the population, in multiple waves. Indeed, for all other populations with evidence of this ancestry, we find much younger admixture dates (Fig. 5a), suggesting that the observed genetic ancestry in north-eastern Europe is inconsistent with a single-pulse admixture event.

**Major genetic shift in Finland since the Iron Age.** Besides the early evidence of Siberian ancestry, our ancient samples from Levänluhta and Chalmny Varre allow us to investigate the more recent population history of Finland. To test whether the ancient individuals from Levänluhta form a clade with modern-day Saami or with modern Finns, we calculated $f_4$(Saami(SGDP), Test; $X$, Mbuti) and $f_4$(Finnish, Test; $X$, Mbuti), where Test was substituted with each ancient individual from Levänluhta, the two historical Saami individuals from Chalmny Varre, as well as the Modern Saami individual, and $X$ was substituted by worldwide modern-day populations (Supplementary Data 5 & 6, and Supplementary Figures 5 & 6). One Levänluhta (JK1968) and the two Chalmny Varre individuals consistently formed a clade with modern-day Saami, but not with modern-day Finns, with respect to all worldwide populations. One Levänluhta individual (JK1970) showed slightly lower affinity to central Europe than modern-day Saami do, while still rejecting a cladal position with modern-day Finns. This indicates that the people inhabiting Levänluhta during the Iron Age, and possibly other areas in the region as well, were more closely related to modern-day Saami than to present-day Finns; however, their difference from the modern Saami may reflect internal structure within the Saami population or additional admixture into the modern population.

One of the individuals from Levänluhta (JK2065/Levänluhta_B) rejects a cladal position with modern Saami to the exclusion of most modern Eurasian populations. This individual also rejects a cladal position with Finns. We analysed low coverage genomes from four additional individuals of the Levänluhta site using PCA (Supplementary Figure 3), confirming the exclusive position of Levänluhta_B compared to all other six individuals (including the four low-coverage individuals) from that site, as is consistent with the ADMIXTURE and qpAdm results. The outlier position of this individual cannot be explained by modern contamination, since it passed several tests for authentication (see Methods) along with all other ancient individuals. However, no direct dating was available for the Levänluhta material, and we cannot exclude the

region so far, and possibly evidence its earliest presence in the western end of the trans-Siberian expanse (Fig. 4). The geographically proximate ancient hunter-gatherers from the Baltics (6000 and 6300 BC) and Motala (~ 6000 BC), who predate Bolshoy, lack this component, as do late Neolithic and Bronze Age individuals from the Baltics[7,8,45]. All later ancient individuals in this study have lower amounts of Nganasan-related ancestry than Bolshoy (Figs 3, 4), probably as a result of dilution through admixture with other populations from further south. This is also consistent with the increased proportion of early European farmer ancestry related to Neolithic Europeans (Fig. 2b) in our later samples. We note that a low but significant amount of Neolithic European ancestry is also present in the Bolshoy population. Finally, we tested whether Bolshoy, instead of Nganasan, can be used as source population. We found that

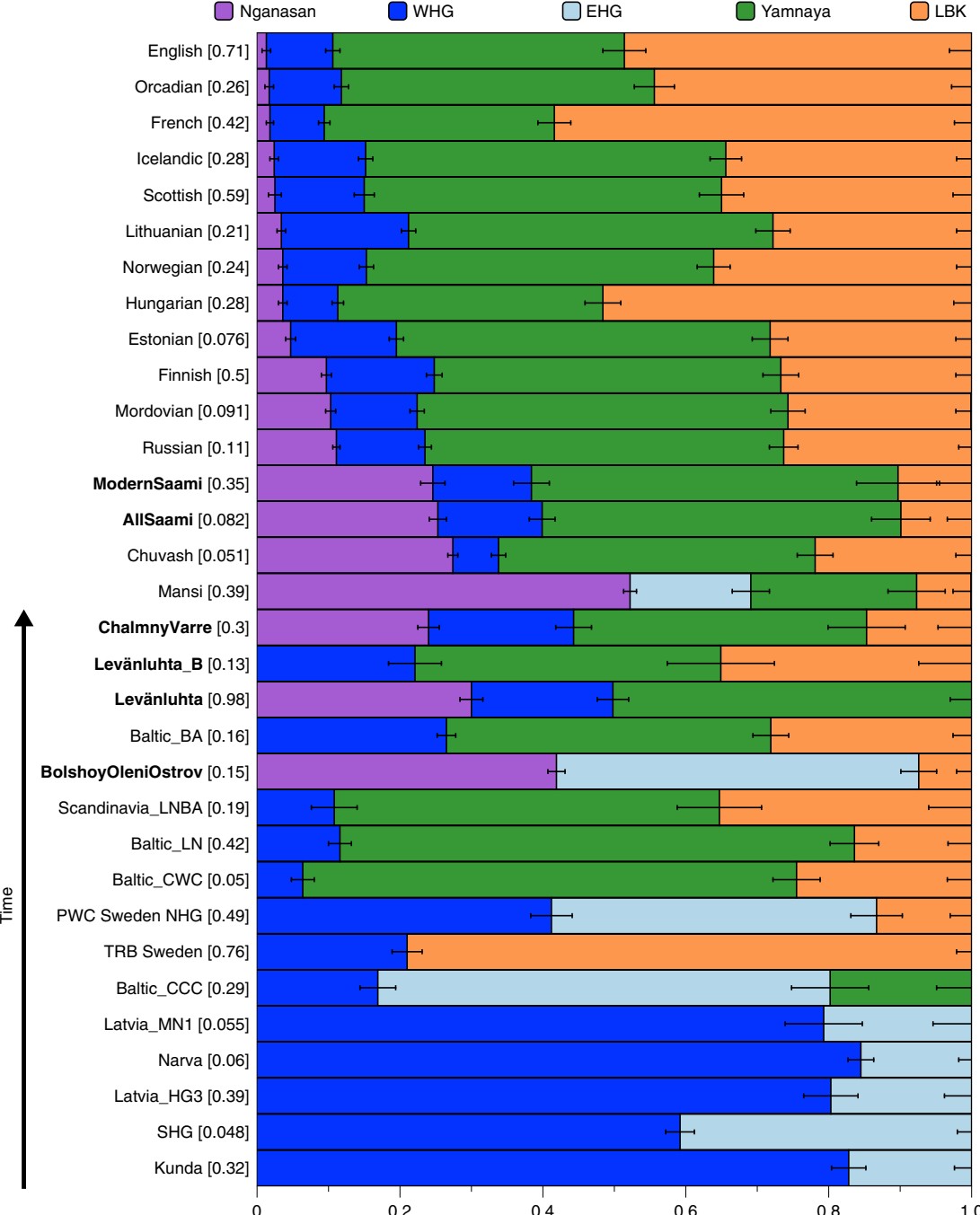

**Fig. 4** Mixture proportions from five sources estimated using qpAdm. Sources used were Nganasan, WHG, EHG, Yamnaya and LBK (see Methods/ Supplementary Data 4). *P*-values (chi-square) for each model are shown in square brackets next to the test population. Results from the least complex model for each test population/individual are shown. "AllSaami" corresponds to a population consisting of the two genomes from the SGDP and the genome from this study (ModernSaami). Error bars represent one standard error and are plotted to the right of their associated mixture proportion. Populations containing individuals from this study are shown in bold. Source data are provided as a Source Data file

possibility of a temporal gap between this individual and the other individuals from that site.

## Discussion

In terms of ancient human DNA, north-eastern Europe has been relatively understudied. In this study we extend the available information from this area considerably, and present the first ancient genome-wide data from Finland. While the Siberian genetic component presented here has been previously described

in modern-day populations from the region[1,3,9,10], we gain further insights into its temporal depth. Our data suggest that this fourth genetic component found in modern-day north-eastern Europeans arrived in the area before 3500 yBP. It was introduced in the population ancestral to Bolshoy Oleni Ostrov individuals 4000 years ago at latest, as illustrated by ALDER dating using the ancient genome-wide data from the Bolshoy samples. The upper bound for the introduction of this component is harder to estimate. The component is absent in the Karelian hunter-gatherers (EHG)[3] dated to 8300–7200 yBP as well as Mesolithic and

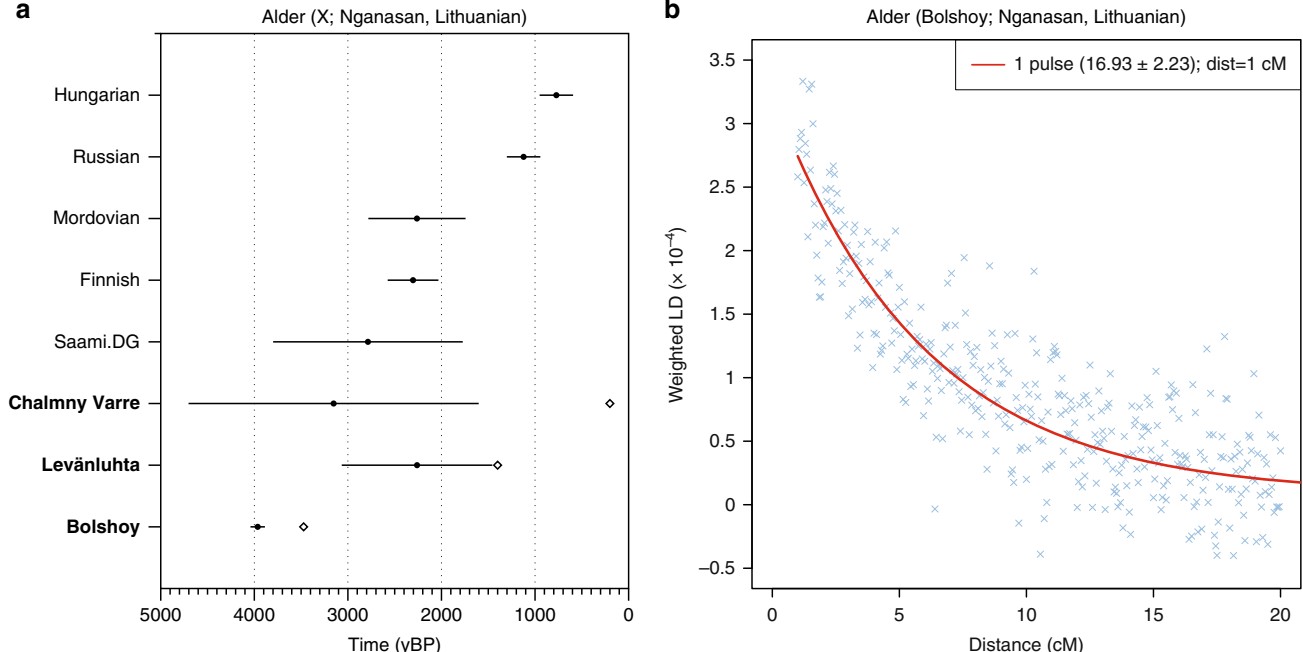

**Fig. 5** Dating the introduction of Siberian ancestry using ALDER. **a** ALDER-inferred admixture dates (filled circles) for different populations, using Nganasan and Lithuanian as sources. Available dates for ancient populations are shown in white diamonds. Error bars represent one standard error provided by ALDER and include the uncertainty surrounding the dating of ancient population samples, calculated using standard propagation. Populations containing individuals from this study are shown in bold. **b** LD decay curve for Bolshoy, using Nganasan and EHG as sources. The fitted trendline considers a minimum distance of 1 cM. A full set of LD decay plots can be found in Supplementary Figure 5. Source data are provided as a Source Data file

Neolithic populations from the Baltics from 8300 yBP and 7100–5000 yBP respectively[8]. While this suggests an upper bound of 5,000 yBP for the arrival of this Siberian ancestry, we cannot exclude the possibility of its presence even earlier, yet restricted to more northern regions, as suggested by its absence in populations in the Baltics during the Bronze Age. Furthermore, our study presents the earliest occurrence of the Y-chromosomal haplogroup N1c in Fennoscandia. N1c is common among modern Uralic speakers, and has also been detected in Hungarian individuals dating to the 10th century[44], yet it is absent in all published Mesolithic genomes from Karelia and the Baltics[3,8,45,49].

The large Nganasan-related component in the Bolshoy individuals from the Kola Peninsula provides the earliest direct genetic evidence for an eastern migration into this region. Such contact is well documented in archaeology, with the introduction of asbestos-mixed Lovozero ceramics during the second millennium BC[50], and the spread of even-based arrowheads in Lapland from 1900 BCE[51,52]. Additionally, the nearest counterparts of Vardøy ceramics, appearing in the area around 1,600-1,300 BCE, can be found on the Taymyr peninsula, much further to the East[51,52]. Finally, the Imiyakhtakhskaya culture from Yakutia spread to the Kola Peninsula during the same period[24,53]. Contacts between Siberia and Europe are also recognised in linguistics. The fact that the Nganasan-related genetic component is consistently shared among Uralic-speaking populations, with the exceptions of absence in Hungarians and presence in the non-Uralic speaking Russians, makes it tempting to equate this genetic component with the spread of Uralic languages in the area. However, such a model may be overly simplistic. First, the presence of the Siberian component on the Kola Peninsula at ca. 3500 yBP predates most linguistic estimates of the spread of extant Uralic languages to the area[54]. Second, as shown in our analyses, the admixture patterns found in historic and modern Uralic speakers are complex and in fact inconsistent with a single admixture event. Therefore, even if the Siberian genetic

component partly spread alongside Uralic languages, some Siberian ancestry may have been already present in the area from earlier admixture events.

The novel genome-wide data presented here from ancient individuals from Finland opens new insights into Finnish population history. Two of the three higher-coverage genomes and all four low-coverage genomes from Levänluhta individuals showed low genetic affinity to modern-day Finnish speakers of the area. Instead, an increased affinity was observed to modern-day Saami speakers, now mostly residing in the north of the Scandinavian Peninsula. These results suggest that the geographic range of the Saami extended further south in the past, and points to a genetic shift at least in the western Finnish region since the Iron Age. The findings are in concordance with the noted linguistic shift from Saami languages to early Finnish. Further ancient DNA from Finland is needed in order to conclude, to what extent these signals of migration and admixture are representative of Finland as a whole.

## Methods

**Sampling**. Written informed consent was obtained from the Saami individual whose genome was analysed in this study, which was approved of by The Hospital District of Helsinki and Uusimaa Ethical Committee (Decision 329/13/03/00/2013) and the ethics committee of the University of Leipzig (Approval reference number 398-13-16122013).

Sampling and extracting ancient DNA requires a strict procedure in order to avoid contamination introduced by contemporary genetic material. For the 13 Iron-Age individuals from Finland available to us, the sampling took place in a clean-room facility dedicated to ancient DNA work, at the Institute for Archaeological Sciences in Tübingen. The preliminary workflow included documenting, photographing and storing the samples in individual, ID-coded plastic tubes and plastic bags. As a result of an early pilot study, the tooth samples we used were fragmented, and some of the dentine was removed. The remaining dentine was collected by carefully separating it from the enamel with a dentist drill and cooled-down diamond drill heads, rotated at a speed below 15 rpm, to avoid possible heat-caused damaging to the ancient DNA.

For samples from the sites of Bolshoy and Chalmny Varre, we used leftover tooth powder that was originally processed at the Institute of Anthropology at the

University of Mainz for replication purposes as described in ref. [24]. In brief, the sample preparation steps included UV-irradiation for 30–45 min, followed by gentle wiping of the surface with diluted commercial bleaches. The teeth were then sand-blasted using aluminium oxide abrasive (Harnisch & Rieth) and ground to fine powder using a mixer-mill (Retsch).

**Radiocarbon date calibration.** We calibrated the radiocarbon date of Bolshoy, reported in refs [24,55] as 3473 ± 42 years BP, using Intcal13[56] as the calibration curve, using OxCal 4.3[57].

**DNA extraction and library preparation.** DNA from the six Bolshoy and the two Chalmny Varre samples was extracted in the ancient DNA facilities of the Max Planck Institute for the Science of Human History (MPI-SHH) in Jena, Germany. Extraction for the Levänluhta samples was similarly conducted in the clean-room facilities of the Institute for Archaeological Sciences in Tübingen. For each specimen, ~ 50 mg of dentine powder was used for an extraction procedure specifically designed for ancient DNA retrieval[58]. Extraction buffer containing 0.45 M EDTA, pH 8.0 (Life Technologies) and 0.25 mg/ml Proteinase K (Sigma-Aldrich) was added to bone powder and incubated at 37 °C with rotation overnight. The supernatant was separated from the pellet of bone powder by centrifugation (14,000 rpm). A binding buffer consisting of 5 M GuHCL (Sigma Aldrich) and 40% Isopropanol (Merck), together with 400 µl of 1 M sodium acetate (pH 5.5) was added to the supernatant, and the solution purified by spinning it through a purification column attached to a High Pure Extender Assembly funnel (8 min. in 1500 rpm, with slow acceleration). The column was then spun into a collection tube (1 min 14,000 rpm) 1–2 times to maximise the yield. This was followed by two subsequent washing steps of 450 µl of wash buffer (High Pure Viral Nucleic Acid Large Volume Kit) and two dry spin steps of 1 min centrifugation at 14,000 rpm. The final total volume of 100 µl eluate was reached by two separate elution rounds of 50 µl of TET (10 mM Tris-HCL, 1 mM EDTA pH 8.0, 0.1% Tween20), each spun for 1 min at 14,000 rpm into a fresh Eppendorf 1.5 ml tube. Negative controls (buffer instead of sample) were processed in parallel at a ratio of 1 control per 7 samples.

Of the 100 µl extract, 20 µl was used to immortalize the sample DNA as a double-stranded library. The procedure included a blunt-end repair, adapter ligation and adapter fill-in steps, as described by Meyer and Kircher[59]. During the blunt-end repair step, a mixture of 0.4 U/µl T4 PNK (polynucleotide kinase) and 0.024 U/µl T4 DNA polymerase, 1× NEB buffer 2 (NEB), 100 µM dNTP mix (Thermo Scientific), 1 mM ATP (NEB) and 0.8 mg/ml BSA (NEB) was added to the template DNA, followed by incubation in a thermocycler (15 min 15 °C, 15 min 25 °C) and purification with a MinElute kit (QIAGEN). The product was eluted in 18 µl TET buffer. The adapter ligation step included a mixture of 1× Quick Ligase Buffer (NEB), 250 nM Illumina Adapters (Sigma-Aldrich) and 0.125 U/µl Quick Ligase (NEB), added to the 18 µl eluate, followed by a 20 min incubation, and second purification step with MinElute columns, this time in 20 µl eluate. For the fill-in step, a mixture of 0.4 U/µl Bst-polymerase and 125 µM dNTP mix was added and the mixture then incubated in a thermocycler (30 min 37 °C, 10 min 80 °C). Libraries without Uracil-DNA-glycosylase (UDG) treatment were produced for all of the 13 extracts from Levänluhta. In addition, UDG-half treated libraries were produced for seven of the original 13 extracts from Levänluhta, and for all Bolshoy and Chalmny Varre extracts. To introduce the UDG-half treatment, an initial stage was included in the library preparation, in which 250 U USER enzyme (NEB) was added into the 20 µl of extract, followed by an incubation at 37 °C for 30 min, and then 12 °C for 1 min. This again was followed by the addition of 200 U UGI (Uracil Glycosylase inhibitor, by NEB) and another identical incubation to stop the enzymatic excision of deaminated sites, as described in[60]. For each library, a unique pair of eight-bp-long indexes was incorporated using a Pfu Turbo Cx Hotstart DNA Polymerase and a thermocycling program with the temperature profile as follows: initial denaturation (98 °C for 30 sec), cycle of denaturation/annealing/elongation (98 °C for 10 sec/ 60 °C for 20 sec/ 72 °C for 20 sec) and final extension at 72 °C for 10 min[61]. Bone powder from a cave bear was processed in parallel serving as a positive control. Negative controls for both extraction and library preparation stages were kept alongside the samples throughout the entire workflow.

Experiment efficiency was ensured by quantifying the concentration of the libraries on qPCR (Roche) using aliquots from libraries before and after indexing. The molecular copy number in pre-indexed libraries ranged from ~ 10E8 to ~ 10E9 copies/µl, indicating a successful library preparation, whereas the indexed libraries ranged from ~ 10E10 to ~ 10E12 copies/µl, stating an admissible indexing efficiency. The negative controls showed 4–5 orders of magnitude lower concentration than the samples, indicating low contamination levels from the laboratory processing stages.

The libraries were amplified with PCR, for the amount of cycles corresponding to the concentrations of the indexed libraries, using AccuPrime Pfx polymerase (5 µl of library template, 2 U AccuPrime Pfx DNA polymerase by Invitrogen, 1 U of readymade 10× PCR mastermix, and 0.3 µM of primers IS5 and IS6, for each 100 µl reaction) with thermal profile of 2 min denaturation at 95 °C, 3–9 cycles consisting of 15 sec denaturation at 95 °C, 30 sec annealing at 60 °C, 2 min elongation at 68 °C and 5 min elongation at 68 °C. The amplified libraries were purified using MinElute spin columns with the standard protocol provided by the manufacturer (Qiagen), and quantified for sequencing using an Agilent 2100 Bioanalyzer DNA 1000 chip.

For the modern Saami individual, total DNA was phenol-chloroform extracted and physically sheared using COVARIS fragmentation. A modified Illumina library preparation was performed using blunt-end repair followed by A-tailing of the 3'-end and ligation of forked adapters. Indexing PCR was followed by excision of fragments ranging from 500 to 600 bp from a 2% agarose gel.

**Capture & sequencing.** We used the in-solution capture procedure from ref. [62] to enrich our libraries for DNA fragments overlapping with 1,237,207 variable positions in the human genome[4]. The sequences used as bait, attached to magnetic beads, were mixed in with the DNA sample template in solution, and left to hybridize with the target DNA during a 24-hour incubation at 60 °C in a rotation oven. 4–6 samples were pooled in equal mass ratios at a total of 2000 ng of DNA. The captured libraries were sequenced (75 bp single-end, plus additional paired-end for the three non-UDG libraries of the Levänluhta individuals) on an Illumina HiSeq 4000 platform at the Max Planck Institute for the Science of Human History in Jena. Out of the 13 originally processed Iron-Age samples from Finland, seven proved to be of an adequate quality to be used in downstream analyses. The modern Saami genome was sequenced in on a Genome Analyser II (8 lanes, 125 bp paired-end) at the Max Planck Institute for Evolutionary Anthropology in Leipzig.

**Processing of sequenced reads.** We used EAGER[63] (version 1.92.50) to process the sequenced reads, using default parameters (see below) for human-originated, UDG-half treated, single-end sequencing data, when processing the UDG-half libraries for all individuals. Specifically, AdapterRemoval was used to trim the sequencing adapters from our reads, with a minimum overlap of 1 base, and using a minimum base quality of 20 and minimum sequence length of 30 bp. BWA aln (version 0.7.12-r1039, https://sourceforge.net/projects/bio-bwa/files)[64] was used to map the reads to the hs37d5 human reference sequence, with a seed length (-l) of 32, max number of differences (-n) of 0.01 while doing no quality filtering. Duplicate removal was carried out using DeDup v0.12.1. Terminal base deamination damage calculation was done using mapDamage, specifying a length (-l) of 100 bp (Supplementary Table 1).

For downstream analyses, we used bamutils (version 1.0.13, https://github.com/statgen/bamUtil.git) TrimBam to trim two bases at the start and end of all reads. This procedure eliminates the positions that are affected by deamination, thus removing genotyping errors that could arise due to ancient DNA damage.

For three Levänluhta individuals exceeding the threshold coverage of 1% in the preliminary screening, we used the non-UDG treated libraries to confirm the authenticity of the ancient data. For these untreated libraries, two rounds of sequencing were carried out, which were processed using EAGER with the above parameters, but specifying a non-UDG treatment mode and setting the correct sequencing type between the libraries. The merged reads were extracted from the resulting bam files, and merged with the bam file containing reads from the single end sequence run using samtools merge (version 1.3)[65].

The modern Saami genome was generated using Ibis for base calling and an in-house adapter trimming script. The resulting reads were then aligned to the hs37d5 human reference genome using bwa 0.5.9-r16 (parameters -e 20 -o 2 -n 0.01).

**Genotyping.** We used a custom program (pileupCaller) to genotype the 15 ancient individuals. A pileup file was generated using samtools mpileup with parameters -q 30 -Q 30 -B containing only sites overlapping with our capture panel. From this file, for each individual and each SNP on the 1240K panel, one read covering the SNP was drawn at random, and a pseudohaploid call was made, i.e., the ancient individual was assumed homozygous for the allele on the randomly drawn read for the SNP in question. PileupCaller is available at https://github.com/stschiff/sequenceTools.git.

For the three Levänluhta libraries that did not undergo UDG treatment, we only genotyped transversions, thus eliminating artefacts of post-mortem C- > T damage from further analyses.

The shotgun genome of the modern Saami individual was genotyped using GATK (version 1.3-25-g32cdef9) Unified Genotyper after indel realignment. The variant calls were filtered for variants with a quality score above 30, and a custom script was used to convert the variants into EigenStrat format.

The data were merged with a large dataset consisting of 3871 ancient and modern individuals genotyped on the Human Origins and/or 1240K SNP arrays, using mergeit.

**Sex determination.** To determine the genetic sex of each ancient individual we calculated the coverage on the autosomes as well as on each sex chromosome. We used a custom script (https://github.com/TCLamnidis/Sex.DetERRmine) for the calculation of each relative coverage as well as their associated error bars (Supplementary Figure 1, Supplementary Note 3 for more information on error calculation). Females are expected to have an x-rate of 1 and a y-rate of 0, while males are expected to have both x- and y-rate of 0.5 (ref. [49]).

**Authentication.** We first confirmed that the deamination pattern at the terminal bases of DNA reads were characteristic of ancient DNA (1.04–4.5% for non-UDG libraries, and 4.7–9.5% for non-UDG libraries, see Supplementary Table 1), using mapDamage (version 2.0.6)[66]. We performed a number of different tests to ensure

the authenticity of our ancient data. For male individuals, we investigated polymorphisms on the X chromosome[27] using the ANGSD software package (version 0.910)[67]. This revealed robust contamination estimates for 2 male Bolshoy individuals, and 1 male Chalmny-Varre individual. All of these were below 1.6% contamination (Table 1). For the female individuals from these two sites, we note that they are projected close to the males in PCA space (Fig. 2a, Supplementary Figure 3), suggesting limited effects of potential contamination. In addition, we generated a PMD-filtered dataset for all individuals using pmdtools (version 0.60)[30]. PMD-filtering was done using a reference genome masked for all positions on the 1240K capture panel, to avoid systematic allelic biases on the analysed SNP positions. We set a pmd-threshold of 3, which, according to the original publication[30], effectively eliminates potential modern contaminants based on the absence of base modifications consistent with deamination.

To provide a more quantitative estimate of possible contamination in females, we used the ContamMix program (version 1.0-10)[29] for estimating mitochondrial contamination. We extracted reads mapping to the mitochondrial reference for each of the ancient individuals using samtools (version 1.3)[65]. We then generated a mitochondrial consensus sequence for each of the ancient individuals using Geneious (version 10.0.9, http://www.geneious.com,[68]), and calling N for all sites with a coverage lower than 5. Finally, all mitochondrial reads were aligned to their respective consensus sequence, using bwa aln (version 0.7.12-r1039)[64] with a maximum number of differences in the seed (-k) set to 5 and the maximum number of differences (-n) to 10, and bwa samse (version 0.7.12-r1039)[64]. A multiple alignment of the consensus sequence and a reference set of 311 mitochondrial genomes[69] was generated, using mafft (version v7.305b)[70–72] with the --auto parameter. The read alignment, as well as the multiple alignment of the consensus and the 311 reference mitochondrial genomes were then provided to ContamMix. We report here a posteriori mode of contamination, along with the upper and lower bounds of the 95% posterior interval (Table 1).

For additional authentication, we ran supervised ADMIXTURE[28] (version 1.3.0) for all samples using the six present-day populations (Atayal, French, Kalash, Karitiana, Mbuti and Papuan) as defined genetic clusters, to locate any large differences in genetic clustering among individuals from the same site (Supplementary Figure 2). We tested the power of this method to detect contamination and find that it can detect contamination that is distantly related to the ancestries present within the test individuals already at rates of 5–8%, but lacks the power to identify contamination closely related to the test individuals (see Supplementary Note 2). We did not observe significant differences (within our resolution) in the ancestry patterns between the ancient individuals from the same site, with the exception of Levänluhta, where the individual sample JK2065 seems to derive from a different ancestry. We therefore assigned it a separate population label, Levänluhta_B in this study.

Finally, using smartpca, we projected PMD-filtered and non-filtered datasets on the same set of principal components constructed on modern European populations, to ensure that the ancient individuals remain projected in the roughly equivalent positions regardless of PMD-filtering. This was possible for all samples with UDG-half treatment, except for the individuals from Levänluhta, which represented too little damage for the PMD-filtering to be applied. Regarding this site, we therefore relied on the non-UDG libraries (using transversions only) that were generated for the three individuals used in the main analysis. We found that within expected noise due to a low number of SNPs, all samples show consistency between the filtered and non-filtered datasets, suggesting a low amount of contamination in all of the samples (Supplementary Figure 3a, b). Four additional individuals from Levänluhta were excluded from the main analysis and from this authentication test because of low coverage (< 15,000 covered SNPs) and lack of non-UDG libraries.

**F statistics**. All programmes used for calculating F statistics in this study can be found as part of the Admixtools package (https://github.com/DReichLab/AdmixTools)[2,42].

We used qp3Pop (version 412) for all $F_3$ calculations.

$F_4$ statistics were calculated using qpDstat (version 711), and qpAdm (version 632)[2] was used to estimate mixture proportions using the following: Sources (Left Populations): Nganasan; WHG; EHG; Yamnaya_Samara; LBK_EN. Outgroups (Right Populations): OG1: Mbuti; CHG; Israel_Natufian; Onge; Villabruna; Ami; Mixe. OG2: Mbuti; CHG; Onge; Villabruna; Ami; Mixe. OG3: Mixe; CHG; Israel_Natufian; Villabruna; Onge; Ami. OG4: Mbuti; Israel_Natufian; Onge; Villabruna; Ami; Mixe. OG5: Mbuti; Samara_HG; CHG; Israel_Natufian; Villabruna; Ami.

To ensure the outgroup sets had enough power to distinguish the ancestries present in the sources, we ran qpWave (version 410) using only the sources as left populations and each outgroup set as rights. All such qpWave runs were consistent only with maximum rank, meaning all outgroup sets had enough power to distinguish between the five different sources. All qpWave and qpAdm models were run using the option allsnps: YES. When the five-way admixture models provided by qpAdm had p-values above 0.05, but included infeasible mixture proportions and one of the sources was assigned a negative mixture proportion, we ran the model again with that source was excluded. For each Test population, if outgroup set OG1 did not produce a working full model (p < 0.05), we tried alternative outgroup sets with one right population removed. This resulted in

outgroup sets OG2-4. In the case of Levänluhta, multiple outgroup sets produced working models, which are listed in Supplementary Data 4. The admixture model needing the minimum number of sources while still providing feasible admixture proportions is always shown. In the case of PWC from Sweden where none of the outgroup sets OG1-4 produced a working model, a revised set of right populations was used (OG5) which includes Samara_HG to provide more power to distinguish hunter-gatherer ancestries. We preferred models with OG1-4 over OG5 in general, because OG5 contains more ancient genomes with potential biases in the right populations, which more often causes failing models for modern Test populations. The excluded sources in the minimal models were specified as N/A (Supplementary Data 4). If either Yamnaya or EHG could be dropped (as is the case for Levänluhta), we show the model which is more consistent with previous publications[3,7,8,45] in Fig. 4, but show both models in Supplementary Data 4.

**Principal component analysis**. We used smartpca (version #16000)[73] (https://github.com/DReichLab/EIG) to carry out Principal Component Analysis (PCA), using the lsqproject: YES and shrinkmode: YES parameters.

For the Eurasian PCA (Fig. 2a), the following populations were used to construct principal components: Abkhasian, Adygei, Albanian, Altaian, Ami, Armenian, Atayal, Avar.SG, Azeri_WGA, Balkar, Balochi, Basque, BedouinA, BedouinB, Belarusian, Borneo, Brahui, Bulgarian, Buryat.SG, Cambodian, Canary_Islanders, Chechen, Chuvash, Croatian, Cypriot, Czech, Dai, Daur, Dolgan, Druze, English, Estonian, Even, Finnish, French, Georgian, Greek, Han, Hazara, Hezhen, Hungarian, Icelandic, Iranian, Italian_North, Italian_South, Japanese, Jew_Ashkenazi, Jew_Georgian, Jew_Iranian, Jew_Iraqi, Jew_Libyan, Jew_Moroccan, Jew_Tunisian, Jew_Turkish, Jew_Yemenite, Jordanian, Kalash, Kalmyk, Kinh, Korean, Kumyk, Kurd_WGA, Kyrgyz, Lahu, Lebanese, Lezgin, Lithuanian, Makrani, Mala, Maltese, Mansi, Miao, Mongola, Mordovian, Naxi, Nganasan, Nogai, North_Ossetian.DG, Norwegian, Orcadian, Oroqen, Palestinian, Pathan, Russian, Saami.DG, Saami_WGA, Sardinian, Saudi, Scottish, Selkup, Semende, She, Sherpa.DG, Sicilian, Spanish, Spanish_North, Syrian, Tajik, Thai, Tibetan.DG, Tu, Tubalar, Tujia, Turkish, Turkmen, Tuvinian, Ukrainian, Ulchi, Uygur, Uzbek, Xibo, Yakut, Yi, Yukagir.

For the West Eurasian PCA (Supplementary Figure 3a, b), the following populations were used to construct principal components: Abkhasian, Adygei, Albanian, Armenian, Balkar, Basque, BedouinA, BedouinB, Belarusian, Bulgarian, Canary_Islander, Chechen, Chuvash, Croatian, Cypriot, Czech, Druze, English, Estonian, Finnish, French, Georgian, Greek, Hungarian, Icelandic, Iranian, Italian_North, Italian_South, Jew_Ashkenazi, Jew_Georgian, Jew_Iranian, Jew_Iraqi, Jew_Libyan, Jew_Moroccan, Jew_Tunisian, Jew_Turkish, Jew_Yemenite, Jordanian, Kumyk, Lebanese, Lezgin, Lithuanian, Maltese, Mordovian, North_Ossetian, Norwegian, Orcadian, Palestinian, Polish, Russian, Sardinian, Saudi, Scottish, Sicilian, Spanish, Spanish_North, Syrian, Turkish, Ukrainian.

**ADMIXTURE analysis**. ADMIXTURE[28] was run with version 1.3.0, following exclusion of variants with minor allele frequency of 0.01 and after LD pruning using plink (version 1.90b3.29)[74] with a window size of 200, a step size of 5 and an $R^2$ threshold of 0.5 (https://www.genetics.ucla.edu/software/admixture/download.html). Five replicates were run for each K value, with K values ranging between 2 and 15. The populations used were: Ami, Ami.DG, Armenian, Atayal, Atayal.DG, Balochi, Basque, BedouinB, Belarusian, Brahmin_Tiwari, Brahui, Chuvash, Croatian, Cypriot, Czech, English, Estonian, Even, Finnish, Finnish.DG, French, Greek, GujaratiB, Hadza, Han, Hungarian, Icelandic, Kalash, Karitiana, Lithuanian, Makrani, Mala, Mansi, Mansi.DG, Mari.SG, Mbuti, Mbuti.DG, Mixe, Mordovian, Nganasan, Norwegian, Onge, Orcadian, Papuan, Pima, Russian, Saami.DG, ModernSaami, Sardinian, Scottish, Selkup, Spanish, Ukrainian, Ulchi, Yoruba, ALPC_Hungary_MN, Baalberge_MN, Baltic_BA, Baltic_CCC, Baltic_CWC, Baltic_LN, BolshoyOleniOstrov, Bu_kk_Culture_Hungary_MN, ChalmnyVarre, CHG, EHG, Esperstedt_MN, Ganj_Dareh_Iran_Neolithic, Hungary_MN, Hungary_Neolithic, Iran_Chalcolithic, JK2065, Koros_Hungary_EN, Kunda, Latvia_HG3, Latvia_MN1, LBK_EN, LBK_Hungary_EN, Levanluhta, Narva, PWC_Sweden_NHG.SG, Scandinavia_LNBA, SHG, Sweden_HG.SG, TRB, Ukraine_HG1, Ukraine_N1, WHG, Yamnaya_Samara.

We find that K = 11 results in the lowest Cross-Validation error, as shown in Supplementary Figure 4b.

**Y-chromosomal haplotyping**. We assigned ancient males to Y haplogroups using the yHaplo program (https://github.com/23andMe/yhaplo)[75]. In short, this program provides an automated search through the Y haplogroup tree (as provided within yHaplo, as accessed from ISOGG on 04 Jan 2016) from the root to the downstream branch based on the presence of derived alleles and assigns the most downstream haplogroup with derived alleles. For about 15,000 Y-chromosomal SNPs present both in our capture panel and in two published datasets[76,77], we randomly sampled a single base and used it as a haploid genotype. We used a custom script to convert EigenStrat genotypes to the yHaplo format. We report the haplogroup assigned furthest downstream provided by the program (Table 1). We also manually checked derived status and absence of mutations defining the designated haplogroup because missing information might lead to a premature stop in its automated search.

**Mitochondrial haplotyping**. We imported the trimmed mitochondrial reads for each individual with mapping quality >30 into Geneious (version 10.0.9, https://www.geneious.com)[68] and reassembled these reads to the reference genome RSRS[78], using the Geneious mapper, with medium sensitivity and 5 iterations. We used the in-built automated variant caller within Geneious to find mitochondrial polymorphisms with a minimum coverage of 3 and a minimum Variant Frequency of 0.67. The resulting variants were exported to Excel and manually compared to the SNPs reported in the online mtDNA phylogeny (mtDNA tree Build 17, 18 Feb 2016, http://www.phylotree.org/). Nucleotide positions 309.1 C(C), 315.1C, AC indels at 515-522, 16182C, 16183C, 16193.1C(C) and 16519 were masked and not included in our haplotype calls.

**Phenotypic SNPs**. We used samtools mpileup (version 1.3)[65], filtering for map- (-Q) and base- (-q) quality of 30, deactivating per-Base Alignment Quality (-B), on the trimmed bam files, to generate a pileup of reads mapping to a set of 43 phenotypic SNPs[4,40,41,79] that are part of our genome capture panel. A custom python script was used to parse the pileup into a table containing the number of reads supporting each allele (Supplementary Data 2).

**Code availability**. All software first described in this study is freely available from online repositories. Sex.DetERRmine: https://github.com/TCLamnidis/Sex.DetERRmine

ContaminateGenotypes: https://github.com/TCLamnidis/ContaminateGenotypes

## Data availability

The raw sequence data of the 16 modern and ancient individuals presented in this paper are deposited at the European Nucleotide Archive (http://www.ebi.ac.uk/ena). The study accession is PRJEB29360. A reporting Summary for this Article is available as a Supplementary Information file. The source data underlying all main and Supplementary Figures are provided as a Source Data file.

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

## Acknowledgements

We thank everyone who contributed to the archaeological excavations. We thank Mikko Putkonen for his notable efforts on early methodological testing and information provided for the Levänluhta samples. We thank Cosimo Posth for carrying out laboratory work, the lab technicians involved in this project, and the sequencing team at the Max Planck Institute for the Science of Human History. We would like to also thank the sequencing team at Max Planck Institute of Evolutionary Anthropology for the sequencing of the modern Saami genome. This project was funded by Emil Aaltonen Foundation, Jane and Aatos Erkko Foundation, the Kone Foundation, Ella and Georg Ehrnrooth Foundation, Jenny and Antti Wihuri Foundation, The Russian State Task for VIGG (AAAA-A16-116111610171-1) and RCMG, the Academy of Finland (grant number: 133056), and the Max Planck Society.

## Author contributions

S.S., J.Kr. and W.H. supervised the study. A.Wes., V.M., V.K., A.S., P.O., O.B., W.H. and J.Kr. assembled the collection of archaeological samples. A.S. and S.P. collected the modern Saami sample. K.M. performed laboratory work. K.M. and T.C.L. supervised ancient DNA sequencing and post-sequencing bioinformatics for the ancient individuals. A.Wei. performed the laboratory work of the modern Saami genome. M.O. carried out the processing of the sequenced reads and generating the genotypes of the modern Saami genome. J.Ke. and S.P. supervised sequencing of the modern Saami and post-sequencing bioinformatics. T.C.L., K.M., E.S., C.J. and S.S. analysed genetic data. T.C.L., K.M., E.S., W.H., J.Kr. and S.S. wrote the manuscript with additional input from all other co-authors.

## Additional information

**Competing interests:** The authors declare no competing interests.

