## [Peer Review File · Nature Communications]

Reviewers' comments:

Reviewer #1 (Remarks to the Author):

Lamnidis, Majander et al present an archaeogenomic study on northeastern European populations. Their samples almost span the last 4000 years, allowing them to see changes over time and compare the patterns seen to neighboring groups. The study has two main results: dating the arrival of Siberian ancestry into the region and observing that groups related to modern Saami may have had a much larger range just a couple of centuries ago. These are really interesting observations which contribute to our understanding of northern European prehistory and, potentially, the spread of Uralic languages.

The authors are using established methodologies, but I do not agree with some of the settings used in their analyses. Furthermore, the authors did not include some published data that would be relevant to their study and might even help to make broader interpretations. Finally, I will also highlight some parts that require a more detailed description in order to make all results reproducible.

=== Major comments ===

I really like the new results about Saami. I would like to add that the ancient samples do not exactly look like the modern-day Saami. So there might have been quite some variation within Saami or that is due to later admixture. I am wondering if there is any information (also from the literature) where modern day Finns came from if they came relatively recently but are more "Siberian" than e.g. Estonians.

The authors make a great effort to estimate contamination by using a range of different approaches. I have a number of comments regarding this:

- The authors do not mention ContamMix in the main text even though it is used for Table 1.
- I think it is confusing to mix the results of two completely different methods in Table 1. Are there confidence intervals for males?
- The observed damage patterns are actually quite minor.
- I am not sure if PMDTools would work with UDG(half) treated libraries – it models an exponential curve for damage along the fragment. Furthermore, the authors say that the original PMDTools publication showed that a threshold of 3 would eliminate contamination. I think that this would have to be verified on a sample basis (e.g. by re-running ContamMix) as it will depend on the sample-specific damage pattern (which doesn't seem to be strong here).
- I don't think the ADMIXTURE approach is suited to estimate low levels of contamination from a somewhat related source.

It is not entirely clear what 511 published ancient samples the authors are using. The authors should provide a supplementary table listing all of them as well as their original publication. They seem to include data from Mittnik et al, which is only available as a preprint at this point so the data is not available to others. That preprint concerns entire northern Europe representing a substantial overlap with the area covered by this study. The authors only seem to be using a very limited subset of the Mittnik et al samples. The title of the present manuscript uses the word "Fennoscandian" which includes the Scandinavian peninsula but the study only uses some Scandinavian hunter-gathers (from ~8000BP) and no other (and younger) samples from the Mittnik et al dataset and other publications (e.g. Skoglund et al 2014). Furthermore, other studies including data from the Baltic states (Jones et al Curr Biol 2016, Saag Curr Biol et al 2017) are not included. Especially the Estonian data from Saag et al could be very interesting for this study: modern Estonians speak a Uralic language, and some of those samples directly predate the arrival of the Siberian component in the new data. That could help

to provide an upper limit for the arrival (at least in the general eastern Baltic and providing that the Estonian samples do not contain the Siberian ancestry). Furthermore, they could provide important context: Saag et al claim that both Yamnaya and Neolithic Anatolian ancestry arrived to Estonia >4500BP while the authors here say that the Siberian ancestry predates Neolithic Anatolian ancestry in the region. This could also be informative with regard to language as a presence of Yamnaya ancestry before the Siberian ancestry would suggest an older Indo-European strata before the Uralic ancestry (and potentially languages) arrive.

The authors base a large part of their conclusions on qpAdm, but I was not able to find a description on how this analysis was done in the Methods section. The only explanations are in the captions to Figure 4 and Supplementary Table 4. Some comments/questions:

- How were the six "right" populations selected (five modern groups, one Upper Palaeolithic)?
- Are the results sensitive to this choice of right populations?
- Are the results shown in Supplementary Table 4 the numbers underlying Figure 4? – If yes, aren't p-values <0.05 interpreted as a rejection of the model, also how should the proportion of Nganasan ancestry be interpreted if other estimates are negative. If no, the authors should describe how the proportions in Figure 4 were derived (maybe subsets of the sources?).

The methods say "...libraries, estimated to pass a threshold of 1% endogenous DNA after capture...". I think the research community would benefit greatly from knowing how that was estimated. Further details/statistics on the sequencing would be informative as well. A supplementary table could include: number of libraries, library complexity, proportion endogenous DNA etc.

I saw in the sequence processing part of the methods that 30 base pairs was used as a length cutoff. I am very concerned about that: Meyer et al. Nature (2016) and Renaud et al Bioinf (2017) showed that such short fragments result in a number of microbial sequences mapping to the human genome. I would recommend to either use only fragments of at least 35bp in length or to show that both thresholds would yield identical results.

The manuscript contains only a single sentence about dating the samples. It is important for the reader to know where the dates in Table 1 come from: Are the samples directly dated (if no how were the dates derived)? Where and how were they dated? Giving the AMS lab codes would be nice as well.

=== Minor comments ===

Please be consistent with the date for the arrival of Siberian ancestry: the abstract uses a conservative 3500 years, while most of the text uses the ALDER estimate of 4000 years.

Please be consistent with Saami vs Sami.

The authors write about "low coverage data" for a modern Saami individual. Please change to SNP array data in order to avoid confusion with "low coverage sequencing".

Most of the analyses are based on 11 individuals mentioned in the abstract, but some include more. I just want to make sure that data for all individuals will be published.

The authors checked their samples for FADS haplotypes which are at high frequencies in Greenland today. Buckley et al MBE (2017) reported some additional sites under selection in modern Europeans and it would be interesting to check those in the ancient samples.

The authors calculate f4 statistics to test what individuals/groups from clades with modern

Finns/Saami. One could directly test f_4 (Saami, Finnish; ancient, Mbuti) for affinity between the ancient "Saami" and either of those modern groups.

I do not agree with this sentence about f_3 statistics: "Additionally, the magnitude of the statistic is directly related to the composition of the tested source populations and how closely those ancestries are related to the actual source populations." My impression is that it rather identifies the extremes along a gradient. Otherwise Gallego-Llorente et al Science (2015) would have detected Sardinian introgression in Africa, and Rodriguez-Varela et al Curr Biol (2017) would have found Scottish or Norwegian introgression on the Canary Islands.

Please always add what individual you are talking about when you discuss the f_4 results ("one Levanluhta individual").

The authors should provide references (or URLs) to the software they have used (e.g. DeDup, bamutils, mergeit, smartpca).

Was smartpca run with the shrinkage parameter? It helps a lot to avoid shrinkage for low coverage samples.

Figure 1 contains site names for some, archaeological cultures for other locations.

Please correct in Materials and Methods, Sampling: "...the specimens were teeth broken down..."

Please correct in Materials and methods, Processing of sequenced reads: "...Sami genome was using Ibis..."

Supplementary Figures 4, 5 and 6 are low resolution or too small to read.

The PCA plot (Figure 2) is quite busy for the reader to understand. Either add some annotation to the plot or be stricter with the colors (e.g. only one color per continental group).

The symbols in Figure 4 could have the same language-based coloring as in Figure 3.

The date for Bolshoy in Figure 5 has a very narrow confidence interval compared to all others. Is that correct? (I'm only surprised since it's a low coverage ancient individual and the uncertainty in age is taken into account.)

Could you provide error bars in Supplementary Figure 1?

Reviewer #2 (Remarks to the Author):

Lamnidis, Majander et al. present new ancient DNA data from the greater Finland area and trace the ancestry of populations in the region over time. In my opinion, this was an interesting and well-executed study, with thorough and appropriate methodology. I only had relatively minor comments, as follows:

I thought that the interpretation of the blue component in Figure 2b, referred to as "The hunter-gatherer genetic ancestry in Europeans" (third paragraph on page 4), was not entirely correct. While it's true that hunter-gatherers have ~100% of the blue component, Yamnaya do as well. Thus, for present-day Europeans, I think that their large blue segments are more likely to reflect primarily

steppe ancestry. If you were to run ADMIXTURE for $k = 4$, would that separate (western) hunter-gatherers from the steppe? I don't think the assignment of the blue component for the majority of Europeans is actually very important for this paper, but it would be interesting to know for northeast Europeans and especially for the ancient samples whether the blue in Figure 2b reflects ancestry that is derived from (eastern) hunter-gatherers or from the steppe. Figure 3b suggests that it is more likely the former, although that statistic is made more complicated by admixture in Yamnaya. I see that the qpAdm results in Supplementary Table 4 also support eastern hunter-gatherer ancestry, albeit with large standard errors for the steppe component (maybe in part due to the close relationship between Karelia and Yamnaya). A bit more discussion of this point could perhaps be interesting.

I found the sentences beginning at "Finally, the present-day Saami as a Test produced the most negative, though non-significant, result..." near the bottom of page 5 to be a little confusing in two respects. First, as currently worded, it maybe sounds as though Saami had the most negative statistic of any test population, which is not what was intended. Second, while it is true that Saami have a large standard error, in fact the magnitude of the f_3 statistic is also barely below 0. Thus, rather than the small sample size and the non-significant Z-score, I think the more relevant point is the weakly negative value (which might be due to larger post-admixture drift in Saami).

The column widths in Table 1 are causing some of the cells to overflow, which makes it a little harder to read.

In Figure 2a, I don't think I can find the marker for JK2065.

In the third paragraph on page 4, I believe the hunter gatherers being used here are more accurately Mesolithic and not Upper Paleolithic. Also I think Loschbour is often grouped with WHG, if you wanted to do that in Figure 2b.

On the x-axis of Figure 4, I would write "Estimated Nganasan-related ancestry" or "Estimated Siberian ancestry."

Ideally it would be good if the resolution could be increased for Supplementary Figure 4.

Three lines from the end of the "Processing of sequenced reads" subsection, there's a typo for "Saami," and I also think there's a word missing in that sentence.

For the paragraph at the top of page 13, while it makes sense in general that individuals from the same site would be expected to cluster together genetically, it might be worth noting that this doesn't necessarily need to be true – and in fact there's significant heterogeneity observed here for Levänluhta.

Reviewer #3 (Remarks to the Author):

I was excited to see this paper in my inbox. The genetic prehistory of Europe has been a topic of considerable interest over the last few years and while much research has focused on western Europe, there are still many unanswered questions about the prehistory of North-East Europe. Of particular intrigue is when and where populations from this region, such as the Finns, picked up Siberian-related ancestry which is not prevalent in Western European populations.

Lamnidis et al., present ancient genome-wide capture data from 11 Iron Age individuals from Finland and North-West Russia. The authors show that even the earliest of these samples (3,500 years old) has the Siberian-related genetic influence which is present in many Northern European populations today. This gives a greater understanding of a poorly characterised aspect of European population genetics. The authors date this admixture to 4,000 years before present. Their well-written results have implications for the ancestral range of Saami populations as well as the spread of Uralic languages.

1. My main concern centres on the discussion of the arrival of Siberian ancestry in Europe. Lamnidis et al., find evidence of Siberian-related ancestry in the earliest sample they analysed dating to 3,500-year-old, which I found convincing. However there is no direct evidence that this is the first arrival of this ancestry in the area. As the authors themselves point out the method used to date this admixture, ALDER, only allows for single pulse admixture and further for multiple pulses it tends to date the most recent admixture event (Moorjani et al., PLOS Genetics, 2011). Hence a date of 4,000 years ago is more informative as a later rather than an early bound. There needs to be more careful discussion of the limitations of this method in the manuscript and statements like “the ADMIXTURE date estimate provided by ALDER for Bolshoy is likely closer to the true time when Siberian ancestry was first introduced in the area” should be revised. I am also unclear about the evidence for the statements “the Bolshoy individuals mark the earliest evidence of Siberian ancestry in north-eastern Europe” and “The component [Siberian] is absent in the Karelian hunter-gatherers (EHG) dated to 8,300-7,200 yBP.” In the qpAdm analysis EHG lack the Siberian component by construction while in ADMIXTURE analysis (Fig 2B) EHG do have a small component of ancestry which defines Siberian (Nganasan) populations. Other studies have shown that EHG are closer to the ancient Siberian sample Mal'ta (MA1) than western hunter-gatherers are (e.g. Haak et al., 2015). Could these results not be due to contacts between Siberia and Europe much earlier than the Iron Age? I would also like to see some alternative models explored using qpAdm, perhaps including MA1 could get around EHG having no Siberian component by design.

2. I liked the methods of assessing the degree of contamination in the ancient samples (f4 and ADMIXTURE approaches) and think that these could be used more widely in the field. I would like to know the sensitivity of these methods. How much contamination can they tolerate? The authors could try to contaminate a modern sample to test this, something like f4(Saami, Saami + X% contamination; contamination source, Chimp) could work. How much contamination does it take to break the clade? A similar approach could be taken with the supervised admixture.

3. In published ADMIXTURE plots Yamnaya are composed of two components - one related to western/eastern hunter-gatherers and the other found in Iranian/Caucasus ancient samples. In the main ADMIXTURE plot of this manuscript (Fig 2B) Yamnaya are almost entirely the same as western hunter-gatherers. This worries me. It would be good for the authors to try running ADMIXTURE with some different populations (maybe add in Caucasus and Iranian samples). I also wonder if the qpAdmix estimates are consistent with other studies (e.g. Haak et al., 2015). The Yamnaya component in French and Hungarian samples seem quite low (about 15%). As said above, it would be beneficial to test different models using qpAdmix rather than presenting just one scenario.

4. I can't locate the sample JK2065 on the main PCA plot. Could the authors make sure it is visible? (I could guess where it should have been from the supp. figure). Also it would be good to plot Uralic speaking populations at the forefront as the authors discuss them throughout the text. I would be interested to know more about the outlying sample JK2065. Is there anything from the archaeology or excavation report which would suggest that this sample is different to the others? Or could it be outlying because of disturbed stratigraphy and the sample actually dates to different period? (“ditching and ploughing” of site described in the supp. materials). This sample should be directly

radiocarbon dated and I would like to see direct radiocarbon dates for the other samples too as much of the discussion centres around dates/arrival times.

5. The authors use admixture f_3 statistics to test populations as mixes of (two) other populations. They found the best results for the Saami modelled this group as a mix of Icelanders and Nganasan. These results are insignificant which is put down to lack of power (low sample size, 3 individuals). It could just as likely be that Icelanders and Nganasan aren't good source populations. To investigate whether power is an issue I would recommend reducing the population size ($n=3$) in some of the other tests presented which yielded significant results to see how sample size affects the results.

Reviewer 1

Reviewer (R): *I really like the new results about Saami. I would like to add that the ancient samples do not exactly look like the modern-day Saami. So there might have been be quite some variation within Saami or that is due to later admixture. I am wondering if there is any information (also from the literature) where modern day Finns came from if they came relatively recently but are more “Siberian” than e.g. Estonians.*

Authors (A): We have expanded on the question about the history of both the Saami and the Finns and have added a few more references on the topic in the Introduction. Indeed, the genetic difference between modern and ancient Saami might also be due to population structure within the Saami population. We have now explicitly mentioned that in the Results section: “This indicates that the people inhabiting Levänluhta during the Iron Age, and possibly other areas in the region as well, were more closely related to modern-day Saami than to present-day Finns; however, their difference from the modern Saami may reflect internal structure within the Saami population or additional admixture into the modern population.”

R: *The authors make a great effort to estimate contamination by using a range of different approaches. I have a number of comments regarding this:*

- The authors do not mention ContamMix in the main text even though it is used for Table 1.

A: Fixed.

R: *- I think it is confusing to mix the results of two completely different methods in Table 1. Are there confidence intervals for males?*

A: We added a column, separated MT from X estimates, and gave confidence intervals for nuclear contamination as well.

R: *- The observed damage patterns are actually quite minor.*

- I am not sure if PMDTools would work with UDG(half) treated libraries – it models an exponential curve for damage along the fragment.

Furthermore, the authors say that the original PMDTools publication showed that a threshold of 3 would eliminate contamination. I think that this would have to be verified on a sample basis (e.g. by re-running ContamMix) as it will depend on the sample-specific damage pattern (which doesn't seem to be strong here).

A: The version of PMDtools that we are using has an updated damage model which takes into account the effects of the uracil-DNA glycosylase, when the respective option is provided (--UDGhalf). Note that the primary data for the analysis was from UDG(half) libraries, so damage is expected to be relatively low. Comparison of PMD-filtered and non-filtered results was only one of many methods to assess contamination. The low nuclear and mitochondrial contamination estimates are also consistent with the results from this analysis. In any case, application of PMDTools results in an enrichment of non-contaminant DNA, and hence any systematic shifts in ancestry between the filtered and non-filtered data would indicate contamination, even if filtering does not remove all contamination.

R: - *I don't think the ADMIXTURE approach is suited to estimate low levels of contamination from a somewhat related source.*

A: This specific test is targeted at distantly related contamination. We added a sentence clarifying this in the main text: "Third, we carried out supervised genetic clustering using ADMIXTURE³¹, using six divergent populations as defined clusters, to identify genetic dissimilarities and possible contamination from distantly related sources (Supplementary Figure 2)"

R: *It is not entirely clear what 511 published ancient samples the authors are using. The authors should provide a supplementary table listing all of them as well as their original publication.*

A: We have now added this information as Supplementary Table 2.

R: *They seem to include data from Mittnik et al, which is only available as a preprint at this point so the data is not available to others. That preprint concerns entire northern Europe representing a substantial overlap with the area covered by this study. The authors only seem to be using a very limited subset of the Mittnik et al samples. The title of the present manuscript uses the word "Fennoscandian" which includes the Scandinavian peninsula but the study only uses some Scandinavian hunter-gathers (from ~8000BP) and no other (and younger) samples from the Mittnik et al dataset and other publications (e.g. Skoglund et al 2014). Furthermore, other studies including data from the Baltic states (Jones et al Curr Biol 2016, Saag Curr Biol et al 2017) are not included. Especially the Estonian data from Saag et al could be very interesting for this study: modern Estonians speak a Uralic language, and some of those samples directly predate the arrival of the Siberian component in the new data. That could help to provide an upper limit for the arrival (at least in the general eastern Baltic and providing that the Estonian samples do not contain the Siberian ancestry). Furthermore, they could provide important context: Saag et al claim that both Yamnaya and Neolithic Anatolian ancestry arrived to Estonia >4500BP while the authors here say that the Siberian ancestry predates Neolithic Anatolian ancestry in the region. This could also be informative with regard to language as a presence of Yamnaya ancestry before the Siberian ancestry would suggest an older Indo-European strata before the Uralic ancestry (and potentially languages) arrive.*

A: We have now incorporated data from Skoglund et al. 2014, Jones et al. 2017, Saag et al. 2017, as well as Mittnik et al. 2018 to our analyses. However, this does not change the uncertainty around an upper limit of the arrival of Siberian ancestry. All of the Mesolithic and Neolithic samples from these additional studies originate thousands of years earlier than the Bolshoy samples. We edited the relevant paragraph around this issue to reflect the newly incorporated data from the Baltics: *“The 3,500-year-old ancient individuals from Bolshoy therefore represent the highest proportion of Siberian ancestry in this region so far, and possibly evidence its earliest presence in the western end of the trans-Siberian expanse (Figure 4). The geographically proximate ancient hunter-gatherers from the Baltics (8,000 and 8,300 BP) and Motala (~8,000 BP), who predate Bolshoy, lack this component. All later ancient individuals in this study have lower amounts of Siberian ancestry than Bolshoy (Figure 3, 4), probably as a result of admixture with other populations from further south and dilution, as also consistent with the increased proportion of early European farmer ancestry related to Neolithic Europeans (Figure 2b) in our later samples.”*

Concerning our claim that Siberian ancestry arrived before Neolithic European ancestry, we now added the following to the discussion: *“The upper bound for the introduction of this component is harder to estimate. The component is absent in the Karelian hunter-gatherers (EHG)3 dated to 8,300-7,200 yBP as well as Mesolithic and Neolithic populations from the Baltics from 8,300 yBP and 7,100-5,000 yBP respectively 8. While this suggests an upper bound of 5,000 yBP for the arrival of Siberian ancestry, we cannot exclude the possibility of its presence even earlier, yet restricted to more northern regions, as suggested by its absence in populations in the Baltic during the Bronze Age”*.

R: *The authors base a large part of their conclusions on qpAdm, but I was not able to find a description on how this analysis was done in the Methods section. The only explanations are in the captions to Figure 4 and Supplementary Table 4. Some comments/questions:*

- *How were the six “right” populations selected (five modern groups, one Upper Palaeolithic)?*
- *Are the results sensitive to this choice of right populations?*
- *Are the results shown in Supplementary Table 4 the numbers underlying Figure 4? – If yes, aren't p-values <0.05 interpreted as a rejection of the model, also how should the proportion of Nganasan ancestry be interpreted if other estimates are negative. If no, the authors should describe how the proportions in Figure 4 were derived (maybe subsets of the sources?).*

A: We have added a section to the Methods clarifying the qpAdm analyses. The outgroups are chosen by putatively differential relationship to the Left populations, in order to give sufficient resolution for discerning the different Left populations. Note that all qpAdm analyses have been redone, using a different outgroup combination resulting in more stable models. We are now reporting the least complex while also feasible model for each test population. Figure 4 has also been updated. Indeed this set of outgroups and sources is suitable in north-eastern but not in central Europe. All qpAdm results are described in the new qpAdm Methods section and new Supplementary Table 4.

R: *The methods say “...libraries, estimated to pass a threshold of 1% endogenous DNA after capture...”. I think the research community would benefit greatly from knowing how that was*

estimated. Further details/statistics on the sequencing would be informative as well. A supplementary table could include: number of libraries, library complexity, proportion endogenous DNA etc.

A: The sentence should read “1% endogenous DNA **before** capture...”. Typo corrected.

R: I saw in the sequence processing part of the methods that 30 base pairs was used as a length cutoff. I am very concerned about that: Meyer et al. Nature (2016) and Renaud et al Bioinf (2017) showed that such short fragments result in a number of microbial sequences mapping to the human genome. I would recommend to either use only fragments of at least 35bp in length or to show that both thresholds would yield identical results.

A: Given our average read lengths, reads with lengths between 30-35 bp make up a small proportion of our total coverage (0.48 - 4.11%). To ensure this proportion was not enough to bias our results, we ran a PC analysis including a version of each ancient individual that only used reads longer than 35bp (figure attached). For most samples, the two points for each sample are indistinguishable. The only visible displacement is with samples JK1963, JK2066, JK2067, which are all at very low coverage, and at even lower coverage with the 35bp threshold, which results in a noisy PCA projection itself

R: The manuscript contains only a single sentence about dating the samples. It is important for the reader to know where the dates in Table 1 come from: Are the samples directly dated (if no how were the dates derived)? Where and how were they dated? Giving the AMS lab codes would be nice as well.

A: Only one direct date is available, which is for Bolshoy, as described in Supplementary Text 1, and its calibration is described in Methods. Unfortunately, an AMS lab code for this

particular date is not available, and we used the date reported in previous publications (see Methods and Supplementary Text 1). Dates for the sites of Levänluhta and Chalmny Varre are context-based, as now summarised in the legend for Table 1.

=== Minor comments ===

R: *Please be consistent with the date for the arrival of Siberian ancestry: the abstract uses a conservative 3500 years, while most of the text uses the ALDER estimate of 4000 years.*

A: The dates are now consistently reported as 3,500 yBP across the manuscript.

R: *Please be consistent with Saami vs Sami.*

A: We now use a consistent naming of “Saami”.

R: *The authors write about “low coverage data” for a modern Saami individual. Please change to SNP array data in order to avoid confusion with “low coverage sequencing”.*

A: The Saami genome is in fact produced by shotgun sequencing. We have made this clearer in the main text.

R: *Most of the analyses are based on 11 individuals mentioned in the abstract, but some include more. I just want to make sure that data for all individuals will be published.*

A: All the data will be available for download upon publication, including the lower coverage genome-wide data. We have also expanded on our sampling, screening and sequencing strategy in the relevant Methods section, and have added the additionally captured low-coverage samples to Table 1.

R: *The authors checked their samples for FADS haplotypes which are at high frequencies in Greenland today. Buckley et al MBE (2017) reported some additional sites under selection in modern Europeans and it would be interesting to check those in the ancient samples.*

A: Sadly, the SNPs from Buckley et al. 2017 do not overlap with the capture panel used here (Mathieson et al. 2015), so we cannot analyse those sites in our data.

R: *The authors calculate f_4 statistics to test what individuals/groups from clades with modern Finns/Saami. One could directly test $f_4(\text{Saami, Finnish; ancient, Mbuti})$ for affinity between the ancient “Saami” and either of those modern groups.*

A: The proposed f_4 test assesses whether each ancient individual has more affinity with Saami or with Finns, instead of testing full cladality with either the Saami or Finnish.

Although that statistic is equivalent to the difference between the two map plots for each ancient individual (Supplementary Figures 5&6), we do not believe this f_4 could act as a replacement for the current statistics, since higher allele sharing with one of the groups does not directly imply cladality with that population.

R: *I do not agree with this sentence about f_3 statistics: “Additionally, the magnitude of the statistic is directly related to the composition of the tested source populations and how closely those ancestries are related to the actual source populations.” My impression is that it rather identifies the extremes along a gradient. Otherwise Gallego-Llorente et al Science (2015) would have detected Sardinian introgression in Africa, and Rodriguez-Varela et al Curr Biol (2017) would have found Scottish or Norwegian introgression on the Canary Islands.*

A: We agree and have removed this sentence and in fact rewritten much of the paragraph containing it. We are now more explicit about what an F_3 statistic can say and how it is affected by genetic drift in the source populations.

R: *Please always add what individual you are talking about when you discuss the f_4 results (“one Levanluhta individual”).*

A: This has been added.

R: *The authors should provide references (or URLs) to the software they have used (e.g. DeDup, bamutils, mergeit, smartpca).*

A: URLs have been added.

R: *Was smartpca run with the shrinkage parameter? It helps a lot to avoid shrinkage for low coverage samples.*

A: Our new PCA plots (Figure 2 & Supp. Figure 3) are done with the “shrinkmode: YES” option in smartpca, as now described in the relevant Methods section.

R: *Figure 1 contains site names for some, archaeological cultures for other locations.*

A: We have updated Figure 1. The previously published sites are now indicated with markers matching those for Figure 2a. Dates and names are only provided for newly reported ancient data to keep the figure readable.

R: *Please correct in Materials and Methods, Sampling: “...the specimens were teeth broken down...”*

A: Corrected.

R: *Please correct in Materials and methods, Processing of sequenced reads: "...Sami genome was using Ibis..."*

A: Corrected.

R: *Supplementary Figures 4, 5 and 6 are low resolution or too small to read.*

A: Our figures and presentation have been updated. They should be easier to read and compare now.

R: *The PCA plot (Figure 2) is quite busy for the reader to understand. Either add some annotation to the plot or be stricter with the colors (e.g. only one color per continental group).*

A: We have edited the figure as suggested, with colors now indicating meaningful groups.

R: *The symbols in Figure 4 could have the same language-based coloring as in Figure 3.*

A: The figure has been updated. We decided to use a barplot instead.

R: *The date for Bolshoy in Figure 5 has a very narrow confidence interval compared to all others. Is that correct? (I'm only surprised since it's a low coverage ancient individual and the uncertainty in age is taken into account.)*

A: "Bolshoy" here refers to the whole population of 5 individuals from the site. The larger sample size (compared to the other ancient populations) is responsible for the smaller error bars. Also, since the Bolshoy individuals are the most recently admixed ones, the LD decay can be estimated much more precisely, which leads to the relatively narrower error bar.

R: *Could you provide error bars in Supplementary Figure 1?*

A: We indeed calculated error bars for the relative coverage on the X and Y chromosomes, as now described in Supplementary Text 2. We also provide a script to calculate these error bars, as described in Supplementary Text 2

Reviewer 2

Reviewer (R): *I thought that the interpretation of the blue component in Figure 2b, referred to as "The hunter-gatherer genetic ancestry in Europeans" (third paragraph on page 4), was not entirely correct. While it's true that hunter-gatherers have ~100% of the blue component,*

Yamnaya do as well. Thus, for present-day Europeans, I think that their large blue segments are more likely to reflect primarily steppe ancestry. If you were to run ADMIXTURE for $k = 4$, would that separate (western) hunter-gatherers from the steppe? I don't think the assignment of the blue component for the majority of Europeans is actually very important for this paper, but it would be interesting to know for northeast Europeans and especially for the ancient samples whether the blue in Figure 2b reflects ancestry that is derived from (eastern) hunter-gatherers or from the steppe. Figure 3b suggests that it is more likely the former, although that statistic is made more complicated by admixture in Yamnaya.

Authors (A): The ADMIXTURE analysis and plot have been updated. The resulting clusters now distinguish Western Hunter gatherers from Steppe ancestry, with Eastern hunter gatherers (EHG) now containing both components.

R: *I see that the qpAdm results in Supplementary Table 4 also support eastern hunter-gatherer ancestry, albeit with large standard errors for the steppe component (maybe in part due to the close relationship between Karelia and Yamnaya). A bit more discussion of this point could perhaps be interesting.*

A: We have updated our qpAdm models. The new Right populations can better distinguish the two components.

R: *I found the sentences beginning at "Finally, the present-day Saami as a Test produced the most negative, though non-significant, result..." near the bottom of page 5 to be a little confusing in two respects. First, as currently worded, it maybe sounds as though Saami had the most negative statistic of any test population, which is not what was intended. Second, while it is true that Saami have a large standard error, in fact the magnitude of the f_3 statistic is also barely below 0. Thus, rather than the small sample size and the non-significant Z-score, I think the more relevant point is the weakly negative value (which might be due to larger post-admixture drift in Saami).*

A: We have now pointed out that post-admixture drift could also confound the result and reworded the sentence accordingly: "This result is still non-significantly negative, either due to the low number of modern Saami individuals in our dataset ($n=3$), or due to post-admixture drift in modern Saami. A high degree of population-specific drift can affect f_3 -statistics and, resulting in less negative and even positive values. This would correlate well with the suggested founder effect in Saami"

R: *The column widths in Table 1 are causing some of the cells to overflow, which makes it a little harder to read.*

A: Table fixed.

R: *In Figure 2a, I don't think I can find the marker for JK2065.*

A: The marker was indeed missing, and we have now added it.

R: *In the third paragraph on page 4, I believe the hunter gatherers being used here are more accurately Mesolithic and not Upper Paleolithic. Also I think Loschbour is often grouped with WHG, if you wanted to do that in Figure 2b.*

A: Corrected.

R: *On the x-axis of Figure 4, I would write “Estimated Nganasan-related ancestry” or “Estimated Siberian ancestry.”*

A: That figure has been changed to a barplot. We still keep “Nganasan” as naming for that component, since that is the actual source used in the model, and “Siberian” is too broad a term in this context.

R: *Ideally it would be good if the resolution could be increased for Supplementary Figure 4.*

A: The figures have been updated; resolution should be better now.

R: *Three lines from the end of the “Processing of sequenced reads” subsection, there’s a typo for “Saami,” and I also think there’s a word missing in that sentence.*

A: Corrected.

R: *For the paragraph at the top of page 13, while it makes sense in general that individuals from the same site would be expected to cluster together genetically, it might be worth noting that this doesn’t necessarily need to be true – and in fact there’s significant heterogeneity observed here for Levänluhta.*

A: Section updated.

Reviewer 3

Reviewer (R): *My main concern centres on the discussion of the arrival of Siberian ancestry in Europe. Lamnidis et al., find evidence of Siberian-related ancestry in the earliest sample they analysed dating to 3,500-year-old, which I found convincing. However there is no direct evidence that this is the first arrival of this ancestry in the area. As the authors themselves point out the method used to date this admixture, ALDER, only allows for single pulse admixture and further for multiple pulses it tends to date the most recent admixture event (Moorjani et al., PLOS Genetics, 2011). Hence a date of 4,000 years ago is more informative as a later rather than an early bound. There needs to be more careful discussion of the*

limitations of this method in the manuscript and statements like “the ADMIXTURE date estimate provided by ALDER for Bolshoy is likely closer to the true time when Siberian ancestry was first introduced in the area” should be revised.

Authors (A): We now have more carefully discussed the ALDER results, including its limitations: *“ALDER provides a relative date estimate for a single-pulse admixture event in generations. When multiple admixture events have occurred, such a single estimate should be interpreted as a (non-arithmetic) average of those events. Therefore the admixture date estimate for Bolshoy does not preclude earlier admixture events bringing Siberian Nganasan-related ancestry into the population, in multiple waves”.*

R: *I am also unclear about the evidence for the statements “the Bolshoy individuals mark the earliest evidence of Siberian ancestry in north-eastern Europe” and “The component [Siberian] is absent in the Karelian hunter-gatherers (EHG) dated to 8,300-7,200 yBP.” In the qpAdm analysis EHG lack the Siberian component by construction while in ADMIXTURE analysis (Fig 2B) EHG do have a small component of ancestry which defines Siberian (Nganasan) populations. Other studies have shown that EHG are closer to the ancient Siberian sample Mal’ta (MA1) than western hunter-gatherers are (e.g. Haak et al., 2015). Could these results not be due to contacts between Siberia and Europe much earlier than the Iron Age? I would also like to see some alternative models explored using qpAdm, perhaps including MA1 could get around EHG having no Siberian component by design.*

A: The mentioned component seen in EHG in the original ADMIXTURE plot (now updated) was consistent with the shared ANE ancestry present in both EHG and Nganasan and thus the results were in fact consistent with no additional Nganasan-related ancestry in EHG. However, we are aware that using EHG as a source in qpAdm renders it by construction as unadmixed. We solved this by explicitly now using the term “Siberian Nganasan-related ancestry” or variations thereof to make clear that the component is by definition distinct from EHG and hence also from ANE or other Siberian ancestral components.

Concerning the time of introduction and the notion of “earliest evidence”, we now rephrased the sentence: *“The 3,500-year-old ancient individuals from Bolshoy represent the highest proportion of Siberian Nganasan-related ancestry seen in this region so far, and possibly evidence its earliest presence in the western end of the trans-Siberian expanse (Figure 4)”.*

R: *I liked the methods of assessing the degree of contamination in the ancient samples (f4 and ADMIXTURE approaches) and think that these could be used more widely in the field. I would like to know the sensitivity of these methods. How much contamination can they tolerate? The authors could try to contaminate a modern sample to test this, something like f4(Saami, Saami + X% contamination; contamination source, Chimp) could work. How much contamination does it take to break the clade? A similar approach could be taken with the supervised admixture.*

A: Unfortunately, further testing has shown that this method is actually too limited in power. We have therefore removed it from the authentication procedure.

R: In published ADMIXTURE plots Yamnaya are composed of two components - one related to western/eastern hunter-gatherers and the other found in Iranian/Caucasus ancient samples. In the main ADMIXTURE plot of this manuscript (Fig 2B) Yamnaya are almost entirely the same as western hunter-gatherers. This worries me. It would be good for the authors to try running ADMIXTURE with some different populations (maybe add in Caucasus and Iranian samples). I also wonder if the qpAdmix estimates are consistent with other studies (e.g. Haak et al., 2015). The Yamnaya component in French and Hungarian samples seem quite low (about 15%). As said above, it would be beneficial to test different models using qpAdmix rather than presenting just one scenario.

A: We have updated the qpAdm and ADMIXTURE analyses, to address these issues.

R: I can't locate the sample JK2065 on the main PCA plot. Could the authors make sure it is visible? (I could guess where it should have been from the supp. figure). Also it would be good to plot Uralic speaking populations at the forefront as the authors discuss them throughout the text. I would be interested to know more about the outlying sample JK2065. Is there anything from the archaeology or excavation report which would suggest that this sample is different to the others? Or could it be outlying because of disturbed stratigraphy and the sample actually dates to different period? ("ditching and ploughing" of site described in the supp. materials). This sample should be directly radiocarbon dated and I would like to see direct radiocarbon dates for the other samples too as much of the discussion centres around dates/arrival times.

A: We have updated the PCA to make Uralic speakers more prevalent and included. The missing marker has been included. Concerning the radiocarbon dates, the amount of material made available to us is unfortunately not sufficient for a direct radiocarbon dating, as pointed out in the Supplementary text for the site. All samples in this study, including the outlier sample JK2065, had consistent special characteristics such as consistency and colour of the material (dark red tint due to a long exposure to the iron-rich water, as opposed to a single separate finding at the site which has been radiocarbon-dated to 19th century). The observed shared features suggest a roughly simultaneous burial date for all the Levänluhta individuals.

R: The authors use admixture f3 statistics to test populations as mixes of (two) other populations. They found the best results for the Saami modelled this group as a mix of Icelanders and Nganasan. These results are insignificant which is put down to lack of power (low sample size, 3 individuals). It could just as likely be that Icelanders and Nganasan aren't good source populations. To investigate whether power is an issue I would recommend reducing the population size (n=3) in some of the other tests presented which yielded significant results to see how sample size affects the results.

A: We have added some more discussion on the f3 results, which mentions a population bottleneck in the Saami, as has been previously suggested. Concerning the significance of the result, admixtools will report a Zscore difference from 0. The expected value of an f3 however is some positive value between 0 and 1. Therefore, a negative value, even if not

significantly different from 0, is consistent with admixture. The significant f_4 results provided for Saami additionally support Siberian admixture into the Saami.

Reviewers' comments:

Reviewer #1 (Remarks to the Author):

I thank the authors for their revisions and for largely addressing my comments. The revised version of the manuscript has improved significantly. I still have some additional comments and I apologize for not raising them earlier.

The qpadm analysis represents a major part for the manuscript and its conclusions. I appreciate the added level of detail to the methods section. Unfortunately, the authors have not addressed all of my comments/questions there. I asked about the p values - my understanding is that low p values are usually interpreted as the model not fitting the data. A substantial number of the models shown in Fig 4 has low p values indeed, so my concern is that they actually do not sufficiently explain the history of these populations.

Additionally, I think the usage of both EHG and Yamnaya (who carry admixture from EHG) as possible sources makes the interpretation of the resulting numbers difficult. EHG ancestry in north-eastern Europe seems to predate any Yamnaya ancestry in the region. The results presented, however, only show Yamnaya ancestry in the later groups and no EHG. If the earlier ("EHG only") groups have contributed any ancestry to the later groups, then those should carry more EHG ancestry than what is represented by Yamnaya. I assume that is the case but qpadm still does not reject the "Yamnaya only" model. The difference between modern Saami and Levänluhta is an example: the text discusses them as closely related (which I believe they are) but in Fig 4 they seem quite different as Levänluhta carry only EHG while Saami only carry Yamnaya ancestry. I assume the explanation is structure within the Saami or that modern Saami have received more recent admixture from Yamnaya-rich groups. The authors should try to disentangle the ancestry components by removing Yamnaya and adding CHG or Iran_Neolithic as source instead (those groups are already in their data set according to supplementary table 2). The presence/absence of CHG/Iran_Neolithic ancestry can still serve as a marker for later Yamnaya-related groups.

I would also like the authors to present supplementary figures with the ADMIXTURE results for all values of K as well as the cross validation errors they are referring to.

Furthermore, I do not think the repeated usage of "outlier" for the genetically different Levänluhta individuals is a good choice (I know this has been done in the aDNA literature before). As a geneticist, I understand the motivation for that - the individual simply seems to originate from a different gene pool, so she should not be grouped with the others for analysis. Using "outlier" too frequently can be interpreted as derogatory, especially by researchers from other disciplines or the general public. She was genetically different, but she was buried with the others so she, as an individual, may have been well integrated and not seen as an outlier. That actually suggests an interesting story for her personally which is independent from the major migrations that us geneticists are usually interested in. I do not really have a good suggestion for alternatives at the moment, the only term I could come up with would be "Levänluhta_west" to represent the western ancestry but that could be confused with a geographic location.

Minor comments:

- Please add the usage of shrinkmode for smartpca to the methods section and not just to figure legends.
- I assume the numbers in Figure 4 represent p values for the models, please add that to the legend.
- Please check the numbers of ancient and modern individuals. Suppl Table 3 contains 3887 in total but the text mentions 3844.

- The methods section is still missing references to publications presenting softwares used for this paper (e.g. bwa, samtools, mafft).
- Supplementary Text 2: Please correct "within out capture panel"

Reviewer #2 (Remarks to the Author):

In my opinion the revised version of the paper does a good job addressing the reviewers' comments, and the new analyses are done well. I only had one small technical question and a few suggestions to explain parts of the figures.

First, I was wondering if the authors could comment on the use of the "allsnps: yes" option in qpAdm. If I'm not mistaken, that means the f4-statistics will be computed using SNPs at which some of the populations in the model (in particular the test population) have no allele call. Is there any risk of bias from having different SNP sets contributing simultaneously to the information in different parts of the matrix?

Second, I thought a few parts of Fig 4 could be clarified. (a) Are the numbers next to the population names p-values (for the model as shown?)? (b) Are the error bars 1 standard error? And does each one only refer to the component to its left?

Likewise, in Fig 5a, are the bars 1 standard error in each direction or 2?

Lastly, I believe the labeling of the Supplementary Table numbers in the Excel files is either off by 1 or should be added for consistency.

Reviewer #3 (Remarks to the Author):

I welcome the authors careful discussion of ALDER dates and longer discussion on the f3 results. However I still have concerns which have not been fully addressed:

The authors repeated the ADMIXTURE analysis with a larger set of populations. Yamnaya still belong to a single "green" cluster. This is at odds with most in not all ancient DNA publications; Haak et al., (2015), Allentoft et al., (2015), Cassidy et al., (2016), Lazaridis et al., (2016), Mittnik et al., (2018) to name but a few. Could the authors provide an explanation for this?

In the ADMIXTURE analysis section the authors state that the American-related ancestry corresponds to an affinity with ANE. Has this been explicitly tested?

In the ADMIXTURE section it is stated that "ANE ancestry also comprises part of the ancestry of Ngeansans". This is not evident from the ADMIXTURE in Figure 2. Where is the evidence for this in this dataset?

The authors found that their f4 contamination test was not sensitive enough to be useful. Did they test their admixture approach too?

The consistency and colour of archaeological material is not a valid dating technique. It is unfortunate

that there is not enough material left to date the outlying JK2065 sample. The possibility of an incorrect date should be discussed in the main text.

Reviewer 1

Reviewer (R): *The qpAdm analysis represents a major part for the manuscript and its conclusions. I appreciate the added level of detail to the methods section. Unfortunately, the authors have not addressed all of my comments/questions there. I asked about the p values - my understanding is that low p values are usually interpreted as the model not fitting the data. A substantial number of the models shown in Fig 4 has low p values indeed, so my concern is that they actually do not sufficiently explain the history of these populations.*

Authors (A): We agree with the reviewer that p-values below 0.05 generally indicate models failing to explain the data fully, which may be a sign for some hidden ancestry component in those populations not explained by the model. We emphasize, though, that our analysis is intended to show a pattern in a relatively large geographic region, which encompasses many populations that may have subtly different ancestry sources. Those differences may bear interest on their own, but are beyond the scope of our analysis here, and do not affect the signal we are mainly describing (the East Asian ancestry in Fenno-Scandian populations).

Nevertheless, we have further revised and improved the qpAdm analyses. First, we have removed populations with failing models in cases clearly outside the scope of this paper. This was the case for Spanish, Ancient and Modern Ukrainians, as well as Bulgarians, who plausibly carry some genetic components from Southern Europe that are not well represented by our source populations attended here mainly for Northern Europe. Second, for the remaining cases with low p-values (e.g. English, French, Hungarian), we have explored modelling with revised outgroup sets. We found we could improve fits substantially by excluding one outgroup from our standard outgroup set, while maintaining the statistical power to distinguish the source populations. In the cases of English, French, Hungarians and Orcadians we found that dropping the Natufian population from the outgroup set puts p-values above 0.05, while maintaining very similar ancestry proportions. In case of Levänluhta, removing Mbuti or Natufians from the outgroups improved the model. In the case of Narva hunter-gatherers from the Baltic, removing CHG from the outgroups also improved the model. Finally, for PWC from Sweden a slightly revised set of 6 outgroups was used (Mbuti, Samara_HG, CHG, Israel_Natufian, Villabruna, Ami), which produced a high p-value. We present in the new main Figure models that mostly have p-values above 0.05 with some borderline exception cases, as discussed in the SI and the main text.

R: *Additionally, I think the usage of both EHG and Yamnaya (who carry admixture from EHG) as possible sources makes the interpretation of the resulting numbers difficult. EHG ancestry in north-eastern Europe seems to predate any Yamnaya ancestry in the region. The results*

presented, however, only show Yamnaya ancestry in the later groups and no EHG. If the earlier (“EHG only”) groups have contributed any ancestry to the later groups, then those should carry more EHG ancestry than what is represented by Yamnaya. I assume that is the case but qpAdm still does not reject the “Yamnaya only” model. The difference between modern Saami and Levänluhta is an example: the text discusses them as closely related (which I believe they are) but in Fig 4 they seem quite different as Levänluhta carry only EHG while Saami only carry Yamnaya ancestry. I assume the explanation is structure within the Saami or that modern Saami have received more recent admixture from Yamnaya-rich groups. The authors should try to disentangle the ancestry components by removing Yamnaya and adding CHG or Iran_Neolithic as source instead (those groups are already in their data set according to supplementary table 2). The presence/absence of CHG/Iran_Neolithic ancestry can still serve as a marker for later Yamnaya-related groups.

A: We believe a deconstruction of Yamnaya ancestry into CHG/Iran vs. EHG is not very useful for modelling most of Europe within the last 2,000 years. In our opinion, a conservative approach for that time window is to use Yamnaya as sole source in general, and only take in EHG if there is specific evidence for extra, or sole, EHG ancestry. In particular, we revised our modelling for Levänluhta (see comments above about our further revised qpAdm-analysis). Depending on the set of outgroups used, we find that the previous model of Levänluhta as a mixture of Nganasan+WHG+EHG+LBK fits the data ($p=0.50$), but so does a mixture model with sources Nganasan+WHG+Yamnaya ($p=0.98$). We now chose the latter of the two models for our main figure, which lacks LBK but is consistent with all other populations from the region after Bolshoy in lacking additional EHG ancestry. We list all models in Supplementary Table 4

R: *I would also like the authors to present supplementary figures with the ADMIXTURE results for all values of K as well as the cross validation errors they are referring to.*

A: We now show ADMIXTURE results for all tried values of K in Supplementary Figure 4. Please note that we have revised the ADMIXTURE analysis (see our first reply to Reviewer 3).

R: *Furthermore, I do not think the repeated usage of “outlier” for the genetically different Levänluhta individuals is a good choice (I know this has been done in the aDNA literature before). As a geneticist, I understand the motivation for that - the individual simply seems to originate from a different gene pool, so she should not be grouped with the others for analysis. Using “outlier” too frequently can be interpreted as derogatory, especially by researchers from other disciplines or the general public. She was genetically different, but she was buried with the others so she, as an individual, may have been well integrated and not seen as an outlier. That actually suggests an interesting story for her personally which is independent from the major migrations that us geneticists are usually interested in. I do not really have a good suggestion for alternatives at the moment, the only term I could come up with would be “Levänluhta_west” to represent the western ancestry but that could be confused with a geographic location.*

A: We appreciate that point, and now suggest simply “Leväluhta” and “Leväluhta_B”, as we also could not come up with more informative names that would not cause confusions due to other reasons. Leväluhta_West indeed suggests a geographic difference, and Leväluhta_Saami wrongly suggests an ethnic association.

R: *Please add the usage of shrinkmode for smartpca to the methods section and not just to figure legends.*

A: Added.

R: *I assume the numbers in Figure 4 represent p values for the models, please add that to the legend.*

A: Added.

R: *Please check the numbers of ancient and modern individuals. Suppl Table 3 contains 3887 in total but the text mentions 3844.*

A: The numbers in the main text are now corrected.

R: *The methods section is still missing references to publications presenting softwares used for this paper (e.g. bwa, samtools, mafft).*

A: Added

R: *Supplementary Text 2: Please correct "within out capture panel"*

A: Done

Reviewer 2

R: *First, I was wondering if the authors could comment on the use of the “allsnps: yes” option in qpAdm. If I’m not mistaken, that means the f4-statistics will be computed using SNPs at which some of the populations in the model (in particular the test population) have no allele call. Is there any risk of bias from having different SNP sets contributing simultaneously to the information in different parts of the matrix?*

A: The allSNPs:YES option indeed means that in each calculated f_4 -statistic, only the four populations contributing need to be covered at a given genomic site, resulting in some f_4 -statistics being applied to different subsets of SNPs. Without the allSNPs options, any SNPs where the Test population is missing a variant call would be excluded from all f_4 calculations. Given that we use a lot of ancient populations in these models (both as sources and outgroups), restricting the analysis only to sites covered by all populations will greatly limit the number of SNPs used and therefore lower the power. In general, therefore, the allSNPs approach leads to increased resolution, for cases where some samples/populations are poorly covered, in particular. This would only cause biased results if there were systematic differences of F_4 statistics across the genome, of which we are not aware. The allSNPs option has been used previously (e.g. Lazaridis et al. 2016).

R: *Second, I thought a few parts of Fig 4 could be clarified. (a) Are the numbers next to the population names p-values (for the model as shown)? (b) Are the error bars 1 standard error? And does each one only refer to the component to its left?*

A: (a) Yes, (b) they are one standard error, and indeed refer to the component on its left. We have added clarifications in fig4 legend.

R: *Likewise, in Fig 5a, are the bars 1 standard error in each direction or 2?*

A: The bars signify one standard error, as well as the uncertainty in C14 dating (where available) propagated using standard error propagation. We have added that to the legend.

R: *Lastly, I believe the labeling of the Supplementary Table numbers in the Excel files is either off by 1 or should be added for consistency.*

A: Sheet names within excel file were incorrectly numbered, and have now been corrected.

Reviewer 3

R: *The authors repeated the ADMIXTURE analysis with a larger set of populations. Yamnaya still belong to a single "green" cluster. This is at odds with most in not all ancient DNA publications; Haak et al., (2015), Allentoft et al., (2015), Cassidy et al., (2016), Lazaridis et al., (2016), Mittnik et al., (2018) to name but a few. Could the authors provide an explanation for this?*

A: ADMIXTURE is a non-deterministic method, and the results randomly differ from run to run, depending also subtly on the exact composition of the input set. The cited studies indeed find consistently that Yamnaya split into two components, but often those components are not EHG + Caucasus (as expected from exact modelling, see Lazaridis et al. 2016), but instead WHG + LBK or WHG + Iran, which is in turn at odds with the standard models for Yamnaya. This reflects the difficulty of resolving these ancestry clusters with ADMIXTURE. The cited studies typically have much larger sample sizes, with up to several thousands of individuals run simultaneously. We have now updated our ADMIXTURE run and added more individuals from South Asia, ancient Iran and Neolithic Europe, increasing the number of analysed individuals from 691 to 926. In our new analysis, Yamnaya is now composed of WHG + Caucasus, consistent with previous publications, (shown in our new Supplementary Figure 4). In this new setup, modern European populations now show genetic components more consistent with previous publications as well.

R: *In the ADMIXTURE analysis section the authors state that the American-related ancestry corresponds to an affinity with ANE. Has this been explicitly tested?*

A: This statement refers simply to the pattern seen in ADMIXTURE, in which Native American cluster components are found in EHG, consistent with previous analyses (Lazaridis et al. 2016). We have added those references also to the main text.

R: *In the ADMIXTURE section it is stated that “ANE ancestry also comprises part of the ancestry of Nganasans”. This is not evident from the ADMIXTURE in Figure 2. Where is the evidence for this in this dataset?*

A: The evidence for ANE ancestry in Nganasans was reported in Lazaridis et al. 2014, which first described the data for them. We have added that reference.

R: *The authors found that their f4 contamination test was not sensitive enough to be useful. Did they test their admixture approach too?*

A: In response to this question, we have added a new analysis to assess the sensitivity of this test, which is summarised in the new Supplementary Text 3. We find that contamination levels of 5% are detectable, if the contamination source is distantly related to the ancestry components in the test samples, and 8% are detectable for more closely related sources. Finally, this approach is not able to detect contamination from sources whose continental ancestry is already present in the test individual. The approach therefore can be used mainly to detect distantly related contamination, for which it was devised originally. We present simulations and a description of the experiments in Supplementary Text 3, and mention the new analysis also in the Methods section on Authentication.

R: The consistency and colour of archaeological material is not a valid dating technique. It is unfortunate that there is not enough material left to date the outlying JK2065 sample. The possibility of an incorrect date should be discussed in the main text.

A: We have discussed this further in the text. The relevant sentence reads:

The outlier position of this individual cannot be explained by modern contamination, since it passed several tests for authentication (see Methods) along with all other ancient individuals. However, no direct dating was available for the Levänluhta material, and we cannot exclude the possibility of a temporal gap between this individual and the other individuals from that site.

REVIEWERS' COMMENTS:

Reviewer #1 (Remarks to the Author):

I thank the authors for their revisions and I recommend the updated version for publication. I only have one minor suggestion: the introduction lists NE European populations as not fitting the 3 source model for most other Europeans. I would suggest to add Estonians and maybe even Hungarians to this list as they also showed an outlier behavior in some of the cited literature and as Uralic speakers they also seem relevant to this study.

Reviewer #2 (Remarks to the Author):

I think the latest version of the paper is very solid, and I have no further comments.

The first referee raised the following point:

I only have one minor suggestion: the introduction lists NE European populations as not fitting the 3 source model for most other Europeans. I would suggest to add Estonians and maybe even Hungarians to this list as they also showed an outlier behavior in some of the cited literature and as Uralic speakers they also seem relevant to this study.

We have added both Estonians and Hungarians in the listed populations in the introduction, since they are indeed specified by Lazaridis et al. 2014 as potentially having additional Asian ancestry. The sentence now reads:

“This model, however, does not fit well for present-day populations from north-eastern Europe such as Saami, Russians, Mordovians, Chuvash, Estonians, Hungarians, and Finns: they carry additional ancestry seen as increased allele sharing with modern East Asian populations^{1,3,9,10}.”